# MAMBATS: IMPROVED SELECTIVE STATE SPACE MODELS FOR LONG-TERM TIME SERIES FORECASTING

## ABSTRACT

In recent years, Transformers have become the de-facto architecture for long-term sequence forecasting (LTSF), yet they face challenges associated with the self-attention mechanism, including quadratic complexity and permutation invariant bias. This raises an important question: *do we truly need the self-attention mechanism to establish long-range dependencies in LTSF?* Recognizing the significance of causal relationships in multivariate LTSF, we propose MambaTS, which leverages causal relationships to model global dependencies across time and variables through a single linear scan. However, causal graphs are often unknown. To address this, we introduce variable-aware scan along time (VAST), which dynamically discovers variable relationships during training and decodes the optimal variable scan order by solving the shortest path visiting all nodes problem during inference. MambaTS employs the latest Mamba model as its backbone. We suggest that the causal convolution in Mamba is unnecessary due to the presence of independent variables, leading to the development of the Temporal Mamba Block (TMB). To mitigate model overfitting, we further incorporate a dropout mechanism for selective parameters in TMB. Extensive experiments conducted on eight public datasets demonstrate that MambaTS achieves new state-of-the-art performance.

## 1 INTRODUCTION

Long-term time series forecasting (LTSF) has a wide range of applications in various fields, including weather, finance, healthcare, energy and transportation (Lim & Zohren, 2021; Wen et al., 2023; Qiu et al., 2024). With the rapid advancement of deep learning, the current methods for time series prediction have shifted from traditional statistical learning approaches to deep learning-based methods, such as recurrent neural networks (RNNs) and temporal convolutional neural networks (TCNs; (Bai et al., 2018; Sen et al., 2019)). Since the introduction of Transformer (Vaswani et al., 2017), Transformer-based methods have emerged as the mainstream LTSF approach (Zhou et al., 2021; Nie et al., 2023), leveraging their self-attention mechanism to effectively capture long-term dependencies in time series data.

Recent studies have identified critical challenges associated with the application of Transformers in LTSF, primarily attributed to the self-attention mechanism inherent in Transformers. One major concern is that the self-attention mechanism often suffers from the curse of quadratic complexity, resulting in a computational cost that escalates rapidly with the context length (Wu et al., 2021; Zhou et al., 2022; Li et al., 2019). Furthermore, recent research has shown that the performance of Transformer-based LTSF methods does not necessarily improve with an increasing look-back window (Nie et al., 2023; Liu et al., 2024b). This phenomenon may be due to the distracted attention caused by the growing input size (Liu et al., 2024b). Additionally, a recent study titled DLinear (Zeng et al., 2023) has challenged the effectiveness of the permutation-invariant bias of the self-attention mechanism in LTSF, achieving remarkable results that surpass those of most state-of-the-art (SOTA) Transformer-based methods using a simple single-layer feed-forward network.

This prompts us to question: *do we truly need the self-attention mechanism to establish long-range dependencies in LTSF?* To address this, we can discuss the issue in two parts. First, regarding temporal dependencies, since time series data is inherently causal, a single linear complexity is sufficient for modeling. Second, when considering variable dependencies, do we require pairwise computations to effectively model these relationships? The answer is no.

Let us establish intuition through a simple example. Consider a causal graph with directed edges representing dependencies among variables, where we aim to predict future outcomes based on these relationships. The causal graph can be represented with edges $A \to C$ and $B \to C$. Due to the nature of causality, a linear scan such as $A, B, C$ or $B, A, C$ effectively captures global dependencies. This is because the information from both $A$ and $B$ inherently contributes to predicting $C$, thus eliminating the need for complex pairwise calculations.

Based on this insight, we propose MambaTS, a novel architecture designed for LTSF that models global dependencies with linear complexity. MambaTS leverages causal graphs to represent the relationships among variables, enabling global dependency modeling through a single linear scan.

However, accurately identifying causal relationships among variables in a dataset poses a significant challenge. To address this, we introduce Variable-Aware Scan along Time (VAST). VAST estimates causal relationships during the training phase using a random walk without return approach, and during the inference phase, it utilizes a shortest path decoding method that visits all nodes (Dréo et al., 2006) to determine the optimal linear scan order.

We utilize the latest Mamba (Gu & Dao, 2023) model as our encoder network. As a formidable competitor to Transformer architectures, Mamba enhances traditional state space model (SSM) (Fu et al., 2023; Gu et al., 2022; Zhang et al., 2023) by introducing selection mechanisms to filter out irrelevant information and reset states along. It also incorporates hardware-aware design for efficient parallel training. Mamba has demonstrated competitive performance compared to Transformers across various domains, offering rapid inference and scalability with sequence length (Dao & Gu, 2024; Zhu et al., 2024; Wang et al., 2024b).

However, we note that the causal convolution in the original Mamba block can be detrimental when scanning independent variables. Consequently, we have removed the local convolution prior to the SSM, leading to the introduction of the Temporal Mamba Block (TMB). Furthermore, previous studies have shown that excessive information integration can lead to overfitting (Nie et al., 2023), a phenomenon we also observed during experiments with MambaTS. To mitigate this, we have added a dropout mechanism (Srivastava et al., 2014) for selective parameters in TMB.

Gained from the insights and designs, MambaTS achieves efficient modeling of global dependencies across time and variables with linear complexity. Extensive experiments on eight popular public datasets demonstrate that MambaTS achieve SOTA performance in most LSTF tasks and settings.

Our contributions are as follows:

1. We introduce MambaTS, a new time series forecasting model based on selective state space models, achieving global dependency modeling through linear scans, supported by theoretical evidence.

2. We propose a method for estimating causal relationships between variables during training via random walk without return, and a decoding strategy based on shortest topological ordering to determine the optimal linear scan sequence during inference.

3. We present the Temporal Mamba Block, which avoids entanglement between independent channels by removing the original causal convolution. Additionally, we incorporate a dropout mechanism for selective parameters in TMB to further prevent overfitting.

4. Our experimental results achieve SOTA performance across various datasets and settings.

## 2 RELATED WORK

**Long-Term Time Series Forecasting.** Traditional LTSF methods leverage the statistical properties and patterns of time series data for prediction (Box & Pierce, 1970). In recent years, LTSF has shifted towards deep learning approaches, where various neural networks are utilized to capture complex patterns and dependencies, elevating LTSF performance. These methods can be broadly classified into two categories: variable-mixing and variable-independent. Variable-mixing methods employ diverse architectures to model dependencies across time and variables. RNNs (Salinas et al., 2020; Shi et al., 2015; Lai et al., 2018) were initially introduced to LTSF due to the nature of sequence modeling. TCNs, known for their local bias, are effective in capturing local patterns in time series data and have shown promising results in LTSF (Bai et al., 2018; Wang et al., 2023; Wu et al., 2023). Transformers

are subsequently introduced to accomplish long-range dependency modeling through self-attention and have become a mainstream method in the Transformer family (Wen et al., 2023). However, due to the quadratic complexity, Transformer-based methods have struggled with optimization efficiency (Zhou et al., 2021; Liu et al., 2022b; Wu et al., 2021; Zhou et al., 2022; Li et al., 2019). Recently, significant improvements have been made in these methods with the introduction of patch-based techniques (Zhang & Yan, 2023; Nie et al., 2023). MLPs are also commonly used for LTSF and have achieved impressive results with their simple and direct architectures (Ekambaram et al., 2023). Graph neural networks have been utilized to model relationships between variables (Wu et al., 2020). FourierGNN (Yi et al., 2023) represents the entire time series information as a hypervariate graph and employs Fourier Graph Neural Network for global dependency modeling. On the other hand, variable-independent methods focus solely on modeling temporal dependencies under the assumption of variable independence (Zeng et al., 2023; Nie et al., 2023; Zhou et al., 2023). These approaches are known for their simplicity and efficiency, often capable of mitigating model overfitting and achieving remarkable outcomes (Zeng et al., 2023; Nie et al., 2023). Nevertheless, this assumption may oversimplify the problem and potentially lead to ill-posed scenarios (Zhang & Yan, 2023).

**State Space Models.** Recently, some works (Box & Pierce, 1970) have combined SSMs (Kalman, 1960) with deep learning and demonstrated significant potential in addressing the long-range dependencies problem. However, the prohibitive computation and memory requirements of state representations often hinder their practical applications (Gu et al., 2022). Several efficient variants of SSMs, such as S4 (Gu et al., 2022), H3 (Fu et al., 2023), Gated State Space (Mehta et al., 2023), and RWKV (Peng et al., 2023), have been proposed to enhance model performance and efficiency in practical tasks. Mamba (Gu & Dao, 2023) addresses a key limitation of traditional SSMs methods by introducing a data-dependent selection mechanism based on S4 to efficiently filter specific inputs and capture long-range context that scales with sequence length. Mamba demonstrates linear-time efficiency in modeling long sequences and surpasses Transformer models in benchmark evaluations (Gu & Dao, 2023).

Mamba has also been successfully extended to non-sequential data such as image (Liu et al., 2024c; Zhu et al., 2024; Wang et al., 2024b), point cloud (Liang et al., 2024), table (Ahamed & Cheng, 2024) and graphs (Wang et al., 2024a; Behrouz & Hashemi, 2024) to enhance its capability in capturing long-range dependencies. To address the scan order sensitivity of Mamba, some studies have introduced bidirectional scanning (Zhu et al., 2024), multi-directional scanning (Li et al., 2024; Liu et al., 2024c), and even automatic direction scanning (Huang et al., 2024). However, there is currently limited work considering the issue of variable scan order in temporal problems. To tackle this challenge, we introduce the VAST strategy to further enhance the expressive power of MambaTS. Similar to our approach, Graph-Mamba (Wang et al., 2024a) also proposes a similar permutation strategy to extend context-aware reasoning on graphs. However, it is based on node prioritization, introducing a biased strategy specifically designed for graphs.

## 3 PRELIMINARIES

**State Space Models.** SSMs (Kalman, 1960) are typically regarded as linear time-invariant (LTI) systems that map continuous input signals $x(t)$ to corresponding outputs $y(t)$ through a state representation $h(t)$. This state space describes the evolution of the state over time and can be represented using ordinary differential equations as follows:

$$h'(t) = \boldsymbol{A}h(t) + \boldsymbol{B}x(t)$$
$$y(t) = \boldsymbol{C}h(t) + \boldsymbol{D}x(t) \tag{1}$$

Here, $h'(t) = \frac{dh(t)}{dt}$, and $\boldsymbol{A}, \boldsymbol{B}, \boldsymbol{C}$, and $\boldsymbol{D}$ are parameters of the time-independent SSMs.

**Discretization.** Finding analytical solutions for SSMs is highly challenging due to their continuous nature. Discretization is typically employed to facilitate analysis and solution in the discrete domain, which involves approximating the continuous-time state space model into a discrete-time representation. This is done by sampling the input signals at fixed time intervals to obtain their discrete-time counterparts. The resulting discrete-time state space model can be represented as:

$$h_k = \overline{\boldsymbol{A}}h_{k-1} + \overline{\boldsymbol{B}}x_k$$
$$y_k = \overline{\boldsymbol{C}}h_k + \overline{\boldsymbol{D}}x_k \tag{2}$$

Here, $h_k$ represents the state vector at time instant $k$, and $x_k$ represents the input vector at time instant $k$. The matrices $\overline{A}$ and $\overline{B}$ are derived from the continuous-time matrices $A$ and $B$ using appropriate discretization techniques such as the Euler or ZOH (Zero-Order Hold) method. In this case, $\overline{\boldsymbol{A}} = \exp(\Delta\boldsymbol{A}), \overline{\boldsymbol{B}} = (\Delta\boldsymbol{A})^{-1}(\exp(\Delta\boldsymbol{A}) - \boldsymbol{I}) \cdot \Delta\boldsymbol{B}$.

**Selective Scan Mechanism.** Mamba further introduces selective SSMs by allowing the parameters to influence the interactions along the sequence in a context-dependent manner. This selective mechanism enables Mamba to filter out irrelevant noise in time series tasks, while selectively propagating or forgetting information relevant to the current input. This differs from previous SSMs methods with static parameters, but it does break the LTI characteristics. Therefore, Mamba takes a hardware optimization approach and implements parallel scan training to address this challenge.

## 4 MOTIVATION

For the multivariate time series forecasting problem, we consider a look-back window of length $L$ for $K$ variables, denoted as $(\mathbf{x}_1, \mathbf{x}_2, \cdots, \mathbf{x}_K)$, where each $\mathbf{x}_i \in \mathbb{R}^L$ represents the values of variable $i$ over the past $L$ time steps. Our objective is to forecast the values of the future $T$ time steps for each variable, denoted as $(\hat{\mathbf{y}}_1, \hat{\mathbf{y}}_2, \cdots, \hat{\mathbf{y}}_K)$, where each $\hat{\mathbf{y}}_i \in \mathbb{R}^T$.

Recent studies (Zeng et al., 2023; Nie et al., 2023) have introduced the variable independence assumption, which reformulates the multivariate LTSF into multiple univariate LTSF. However, this assumption can be overly simplistic and may lead to ill-posed scenarios (Zhang & Yan, 2023). In practical applications, multivariate time series often exhibit causal relationships; neglecting these dependencies can result in suboptimal predictive performance. Therefore, modeling variable dependencies remains a dominant approach, particularly exemplified by Transformer-based methods that operate with quadratic complexity in pairwise computations (Liu et al., 2024b). Some researches, such as Zhang & Yan (2023), seek to mitigate this complexity through a router mechanism that aggregates information from all variables prior to redistribution. Nonetheless, these methods still struggle with efficiency and scalability.

Given these challenges, we pose the question: *Is it possible to achieve global dependency modeling with linear complexity without sacrificing the integrity of variable dependencies?* The answer lies in leveraging the structural properties of causal relationships among variables. By representing these relationships through causal graphs, we can effectively capture the fundamental dependencies that drive multivariate interactions. The following theorem formalizes this approach:

**Proposition 1.** *For a multivariate time series dataset, if a causal graph $G = (V, E)$ exists that represents the relationships among the variables $\mathbf{V} = \{\mathbf{V}_1, \mathbf{V}_2, \ldots, \mathbf{V}_K\}$, then global dependency modeling can be achieved through a single linear scan.*

*Proof.* A causal graph $G$ is constructed, where $V$ represents the variables and $E$ indicates the causal relationships. Since $G$ is a directed acyclic graph (DAG), there exists at least one topological order $\sigma$ such that for any directed edge $(\mathbf{V}_i, \mathbf{V}_j)$, $\mathbf{V}_i$ precedes $\mathbf{V}_j$ in this order. Following this topological ordering during a linear scan, each variable $\mathbf{V}_{\sigma(k)}$ can be conditioned on its parents $\mathrm{Pa}(\mathbf{V}_{\sigma(k)})$, allowing us to represent dependencies as $P(\mathbf{V}_{\sigma(k)}|\mathrm{Pa}(\mathbf{V}_{\sigma(k)}))$. Consequently, the joint distribution of all variables is obtained as $P(\mathbf{V}) = \prod_{i=1}^{K} P(\mathbf{V}_{\sigma(i)}|\mathrm{Pa}(\mathbf{V}_{\sigma(i)}))$. This process demonstrates that a single linear scan effectively models global dependencies in multivariate time series by leveraging the structural information encoded in the causal graph. $\square$

Despite Proposition 1 demonstrating the potential for global dependency modeling with known causal relationships, real-world datasets often lack explicit dependency among variables. As a result, the underlying causal structure can be obscured by data complexity and interactions. To address this, it is essential to adopt methods for inferring causal relationships without a predefined graph. A promising approach is the random walk (Przymus et al., 2017; Kim et al.), which employs stochastic processes to explore the graph's structure and estimate causal links based on empirical transition observations.

**Definition 1** (**Random Walk Without Return**). *A random walk without return is a stochastic process defined on a graph where a walker starts at a node and moves to an adjacent node uniformly at random, with the restriction that it cannot return to previously visited nodes until all nodes have been visited.*

**Proposition 2.** *Given a causal graph $G = (V, E)$ with unknown relationships among nodes $\mathbf{V} = \{\mathbf{V}_1, \mathbf{V}_2, \ldots, \mathbf{V}_K\}$, if the total cost of a random walk without return is known, then the causal relationships can be estimated through infinite random walks without return.*

*Proof.* In a random walk without return, the total cost $C$ is shared across $K - 1$ transitions. Assuming that each transition contributes evenly to $C$, we focus on whether $C$ accurately reflects the true transition cost. Transitions are classified into three types: positive transitions (PT), where $C_{i,j}^{\text{PT}} > 0$ if $\mathbf{V}_i \rightarrow \mathbf{V}_j$ is a causal relationship; negative transitions (NT), where $C_{i,j}^{\text{NT}} = -C_{j,i}^{\text{PT}}$ if $\mathbf{V}_j \rightarrow \mathbf{V}_i$ is causal; and independent node transitions (IN), where $0 < C_{i,j}^{\text{IN}} < C_{i,j}^{\text{PT}}$ if no causal relationship exists. Due to the symmetry of the graph, $\#\text{PT} = \#\text{NT}$, and thus at least $\frac{\#\text{PT}+\#\text{IN}}{\#\text{PT}+\#\text{NT}+\#\text{IN}} \geq \frac{1}{2}$ of the transitions contribute to the cost update, with equality if and only if $\#\text{IN} = 0$. As $N \rightarrow \infty$, the expected cost for each transition converges to a positive value, $\mathbb{E}[C_{i,j}^{(n)}] > 0$, and the average cost after $N$ random walks is $p_{i,j} = \frac{1}{N} \sum_{n=1}^{N} C_{i,j}^{(n)}$. Finally, the strength of the causal relationship between $\mathbf{V}_i$ and $\mathbf{V}_j$ is estimated as $\hat{R}_{i,j} = \frac{p_{i,j}}{\sum_{k \in V} p_{i,k}}$, allowing causal relationships to be accurately estimated as $N \rightarrow \infty$. A full proof can be found in Appendix A. $\square$

Through Proposition 1 and 2, we establish that for any multivariate dataset that can be represented by a causal graph, global dependency modeling can be efficiently accomplished with linear complexity. In the next section, we will present a detailed instantiation of this framework, termed MambaTS, which integrates these theoretical underpinnings into a practical methodology for effective LTSF.

## 5 METHODOLOGY

### 5.1 OVERALL ARCHITECTURE

For clarity, We assume that the order of the $K$ variables satisfies the linear scanning condition, and we will present the estimation of variable order relationship in Section 5.2. The architecture of MambaTS is illustrated in Figure 1. It primarily consists of an embedding layer, an instance normalization layer, $N \times$ Temporal Mamba blocks, and a prediction head.

**Patching and Embedding** As shown in Figure 1 (a), for the input $\mathbf{x} = (\mathbf{x}_1, \mathbf{x}_2, \cdots, \mathbf{x}_K) \in \mathbb{R}^{K \times L}$, we adopt the segmentation approach used in PatchTST, dividing each variable into patches every $s$ time steps. This process yields $M = \lceil \frac{L}{s} \rceil$ patches for each variable, where $\mathbf{x}_k = (\mathbf{x}_k^{(1)}, \mathbf{x}_k^{(2)}, \cdots, \mathbf{x}_k^{(M)})$ for $k = 1, 2, \ldots, K$. These patches are then embedded into $D$-dimensional tokens via a linear mapping defined as $\mathbf{z}_k^{(j)} = \mathbf{W} \cdot \mathbf{x}_k^{(j)}$, where $\mathbf{W} \in \mathbb{R}^{D \times s}$ is the weight matrix.

**Variable Scan along Time** By embedding $K$ variables, we obtain $K \times M$ tokens, which we then interleave into the structure $\mathbf{z} = (\mathbf{z}_1^{(1)}, \mathbf{z}_2^{(1)}, \ldots, \mathbf{z}_1^{(2)}, \mathbf{z}_2^{(2)}, \ldots, \mathbf{z}_{K-1}^{(M)}, \mathbf{z}_K^{(M)})$, referred to as Variable Scan along Time (VST). Unlike the single-variable temporal organization of PatchTST (Nie et al., 2023) or the variable-level arrangement of iTransformer (Liu et al., 2024b), VST adopts an organized format that enables a fine-grained and effective representation of retrospective information. The resulting VST tokens are then input into the encoder to model global dependencies across both time and variables.

**Encoder** The encoder consists of $N$ stacked Temporal Mamba Block (TMB), modified from the Mamba Block (Gu & Dao, 2023). The architecture of the Mamba Block is illustrated in Figure 1 (b) and features two branches: the right branch focuses on sequence modeling, while the left contains a gated non-linear layer. The computation process of the original Mamba Block is defined as follows:

$$\mathbf{h} = \text{SSM}(\text{Conv}(\text{Linear}(\mathbf{z}))) + \sigma(\text{Linear}(\mathbf{z})). \tag{3}$$

Here, the causal convolution Conv, acting as a shift-SSM, is inserted before the main SSM layer to enhance connections between adjacent tokens. However, given that the $K$ variables may exhibit independence, the TMB (see Figure 1 (c)) removes this component. Additionally, the TMB introduce a dropout mechanism for selective parameters to mitigate overfitting, as detailed below:

$$\mathbf{h} = \text{SSM}(\text{Dropout}(\text{Linear}(\mathbf{z}))) + \sigma(\text{Linear}(\mathbf{z})). \tag{4}$$

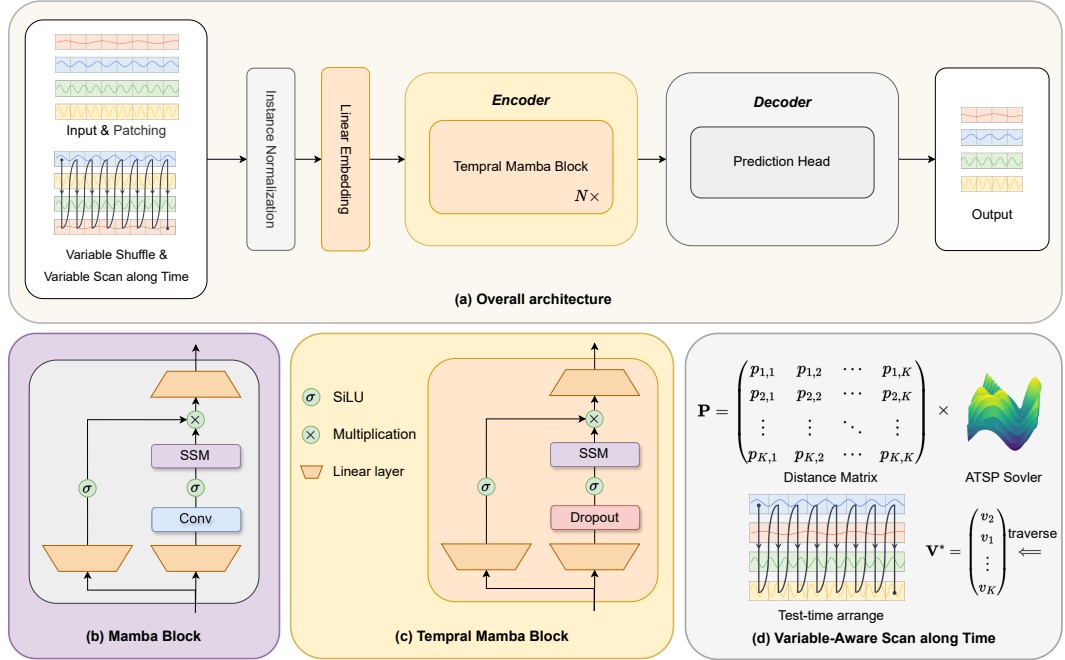

Figure 1: The overall architecture of MambaTS.

**Prediction Head**   During the decoding phase, akin to PatchTST (Nie et al., 2023), we predict independently for each variable. This process involves a simple linear head that transforms the encoded result $\mathbf{h} = (\mathbf{h}_1, \mathbf{h}_2, \cdots, \mathbf{h}_K) \in \mathbb{R}^{K \times (MD)}$ to $\hat{\mathbf{y}} \in \mathbb{R}^{K \times T}$. This setup ensures the model to generate forecast results effectively.

**Instance Normalization**   To mitigate distribution shift between training and test data, following RevIN (Kim et al., 2022), we standardize each input channel to zero mean and unit standard deviation, and retain these statistics for the de-normalization of model predictions.

**Loss Function**   We adopt mean squared error (MSE) loss as the primary loss function, given by:

$$\mathcal{L} = \frac{1}{K} \sum_{i=1}^{K} \|\hat{\mathbf{y}}_i - \mathbf{y}_i\|_2^2. \tag{5}$$

## 5.2   VARIABLE-AWARE SCAN ALONG TIME

As emphasized in Proposition 1, the sequence of variable scanning is essential for accurately modeling global variable dependencies. However, the relationships among variables are not known a priori. Motivated by Proposition 2, we propose Variable-Aware Scan along Time (VAST). The core idea of VAST is to estimate causal relationships among variables during the training phase through random walks without return, while during inference, it determines the optimal linear scanning order using the shortest paths that traverse all nodes.

**Training.**   In the training phase, we introduce the variable permutation training (VPT) strategy. Specifically, for the input $\mathbf{x} = (\mathbf{x}_1, \mathbf{x}_2, \cdots, \mathbf{x}_K) \in \mathbb{R}^{K \times L}$, we randomly shuffle the order of variables in each iteration to perform a random walk without return. We then revert the shuffled state after decoding to ensure the correct output sequence.

For any $K$ variables, we maintain a directed graph adjacency matrix $\boldsymbol{P} \in \mathbb{R}^{K \times K}$, where $p_{i,j}$ represents the cost from node $i$ to node $j$. Notably, with the introduction of VPT, we can explore various combinations of scan orders and evaluate their value using Eq. 5. Following a permutation, a node index sequence $\mathbf{V} = \{v_1, v_2, \cdots, v_K\}$ is obtained, where $v_k$ denotes the new index in

the shuffled sequence. Subsequently, $K - 1$ transition tuples $\{(v_1, v_2), (v_2, v_3), \cdots (v_{K-1}, v_K)\}$ are derived. For each sample, a training loss $l^{(t)}$ is generated in the $t$-th iteration of the network. Therefore, we update $\boldsymbol{P}$ with exponential moving average:

$$p_{v_k, v_{k+1}}^{(t)} = \beta p_{v_k, v_{k+1}}^{(t-1)} + (1 - \beta) l^{(t)}, \tag{6}$$

where $\beta$ is a hyperparameter controlling the sliding average rate, determining the impact of new estimates on the global variable importance. To facilitate efficient training, we extend the above formula to a batch version. By a simple nondimensionalization operation to eliminate the influence of different sample batches, we define $\bar{\mathbf{l}}^{(t)} = \mathbf{l}^{(t)} - \text{dev}(\mathbf{l}^{(t)})$, where dev denotes the standard deviation and $\mathbf{l}^{(t)}$ refers to the batch version of $l^{(t)}$. Consequently, the Eq. 6 is modified to:

$$\boldsymbol{P}^{(t)} = \beta \boldsymbol{P}^{(t-1)} + (1 - \beta) \bar{\mathbf{l}}^{(t)}. \tag{7}$$

**Inference** Throughout training, $\boldsymbol{P}$ are leveraged to determine the optimal variable scan order. This involves solving the asymmetric traveling salesman problem (ATSP), which seeks the shortest path visiting all nodes. Given the dense connectivity represented by $\boldsymbol{P}$, finding the optimal traversal path is NP-hard. Hence, we introduce a heuristic-based simulated annealing (Dréo et al., 2006) algorithm for path decoding.

# 6 EXPERIMENTS

Table 1: Summary of Dataset Characteristics

| Datasets | ETTh2 | ETTm2 | Weather | Electricity | Traffic | Solar | Covid-19 | PEMS |
|---|---|---|---|---|---|---|---|---|
| Features | 7 | 7 | 21 | 321 | 862 | 137 | 948 | 358 |
| Time steps | 17,420 | 17,420 | 52,696 | 26,304 | 17,544 | 52,179 | 1,392 | 21,351 |
| Frequency | 1 hour | 15 mins | 10 mins | 1 hour | 1 hour | 10 mins | 1 day | 5 mins |

**Dataset.** We conducted extensive experiments on eight public datasets, as shown in Table 1, including two ETT datasets (Zhou et al., 2021), Weather, Electricity, Traffic (Wu et al., 2021), Solar (Lai et al., 2018), Covid-19 (Panagopoulos et al., 2021), and PEMS (Liu et al., 2022a), covering domains such as electricity, energy, transportation, weather, and health.

**Baselines and Metrics.** To demonstrate the effectiveness of MambaTS, we compared it against SOTA models of LTSF, including five popular Transformer-based methods: PatchTST (Nie et al., 2023), iTransformer (Liu et al., 2024b), FEDformer (Zhou et al., 2022), Autoformer (Wu et al., 2021), and three competitive non-Transformer-based methods: DLinear (Zeng et al., 2023), MICN (Wang et al., 2023), and FourierGNN (Yi et al., 2023). Following PatchTST (Nie et al., 2023), we primarily evaluate the models using Mean Squared Error (MSE) and Mean Absolute Error (MAE).

**Implementation Details.** Experiments were performed on an NVIDIA RTX 3090 Ti 24 GB GPU using the Adam optimizer (Kingma & Ba, 2015) with betas of (0.9, 0.999). Training ran for 10 epochs with early stopping implemented using a patience of 3 to avoid overfitting. The best parameter selection for all comparison models was carefully tuned on the validation set.

## 6.1 MAIN RESULTS

Table 2 displays the results of multivariate long-term forecasting. Overall, MambaTS achieved new SOTA results (highlighted in red bold) across various prediction horizons on most datasets. While DLinear and PatchTST assume variable independence and perform well on datasets with a small number of variables like ETTh2/m2 ($K = 7$) and Weather ($K = 21$), their performance diminishes on complex datasets with a larger number of variables such as Traffic ($K = 862$) and Covid-19 ($K = 948$), highlighting the limitations of the variable-independent assumption. Conversely, iTransformer exhibited contrasting performance, excelling on intricate datasets but underperforming on datasets with fewer variables. Other baselines demonstrated competitive results on specific datasets under certain prediction scenarios.

Table 2: Multivariate long-term series forecasting results. All models employ a look-back window length of $L = 96$ for the Covid-19 dataset and $L = 720$ for the remaining datasets.

| Models | | MambaTS (ours) | | iTransformer (2024) | | FourierGNN (2024) | | PatchTST (2023) | | Dlinear (2023) | | MICN (2023) | | FEDformer (2022) | | Autoformer (2021) | |
|---|---|---|---|---|---|---|---|---|---|---|---|---|---|---|---|---|---|
| Metric | | MSE | MAE | MSE | MAE | MSE | MAE | MSE | MAE | MSE | MAE | MSE | MAE | MSE | MAE | MSE | MAE |
| ETTh2 | 96 | **0.281** | **0.347** | 0.312 | 0.363 | 0.454 | 0.481 | 0.283 | **0.347** | 0.306 | 0.370 | 0.289 | 0.354 | 0.332 | 0.374 | 0.332 | 0.368 |
| | 192 | **0.352** | 0.397 | 0.384 | 0.408 | 0.560 | 0.541 | 0.354 | **0.391** | 0.411 | 0.437 | 0.408 | 0.444 | 0.407 | 0.446 | 0.426 | 0.434 |
| | 336 | **0.372** | **0.416** | 0.431 | 0.444 | 0.608 | 0.568 | 0.376 | **0.411** | 0.542 | 0.514 | 0.547 | 0.516 | 0.400 | 0.447 | 0.477 | 0.479 |
| | 720 | 0.404 | 0.444 | 0.432 | 0.463 | 0.820 | 0.648 | **0.402** | **0.440** | 0.900 | 0.671 | 0.834 | 0.688 | 0.412 | 0.469 | 0.453 | 0.490 |
| ETTm2 | 96 | 0.172 | 0.263 | 0.181 | 0.275 | 0.229 | 0.327 | 0.168 | 0.259 | **0.163** | **0.258** | 0.177 | 0.274 | 0.180 | 0.271 | 0.205 | 0.293 |
| | 192 | 0.228 | **0.304** | 0.243 | 0.315 | 0.308 | 0.384 | 0.237 | 0.309 | **0.222** | **0.304** | 0.252 | 0.310 | 0.252 | 0.318 | 0.278 | 0.336 |
| | 336 | 0.284 | 0.343 | 0.297 | 0.352 | 0.362 | 0.413 | 0.279 | **0.336** | **0.274** | **0.336** | 0.299 | 0.350 | 0.324 | 0.364 | 0.343 | 0.379 |
| | 720 | **0.356** | 0.392 | 0.381 | 0.404 | 0.482 | 0.487 | 0.363 | **0.390** | 0.407 | 0.432 | 0.421 | 0.434 | 0.410 | 0.420 | 0.414 | 0.419 |
| Weather | 96 | **0.145** | **0.195** | 0.180 | 0.232 | 0.162 | 0.232 | 0.149 | 0.198 | 0.168 | 0.227 | 0.167 | 0.231 | 0.238 | 0.314 | 0.249 | 0.329 |
| | 192 | **0.192** | **0.241** | 0.228 | 0.270 | 0.207 | 0.276 | 0.194 | **0.241** | 0.212 | 0.267 | 0.212 | 0.271 | 0.275 | 0.329 | 0.325 | 0.370 |
| | 336 | **0.245** | **0.283** | 0.291 | 0.316 | 0.261 | 0.318 | **0.245** | 0.282 | 0.256 | 0.305 | 0.275 | 0.337 | 0.339 | 0.377 | 0.351 | 0.391 |
| | 720 | 0.314 | **0.330** | 0.354 | 0.359 | 0.336 | 0.366 | 0.314 | 0.334 | 0.315 | 0.355 | **0.312** | 0.349 | 0.389 | 0.409 | 0.415 | 0.426 |
| Electricity | 96 | **0.128** | **0.223** | 0.130 | **0.223** | 0.133 | 0.229 | 0.133 | 0.230 | 0.151 | 0.260 | 0.166 | 0.274 | 0.186 | 0.302 | 0.196 | 0.313 |
| | 192 | **0.146** | **0.239** | 0.147 | 0.240 | 0.155 | 0.251 | 0.147 | 0.244 | 0.165 | 0.276 | 0.182 | 0.289 | 0.197 | 0.311 | 0.211 | 0.324 |
| | 336 | **0.161** | 0.258 | 0.164 | **0.257** | 0.167 | 0.264 | 0.162 | 0.261 | 0.183 | 0.291 | 0.201 | 0.308 | 0.213 | 0.328 | 0.214 | 0.327 |
| | 720 | **0.187** | **0.283** | 0.203 | 0.292 | 0.194 | 0.288 | 0.196 | 0.294 | 0.201 | 0.312 | 0.235 | 0.339 | 0.233 | 0.344 | 0.236 | 0.342 |
| Traffic | 96 | **0.347** | **0.248** | 0.349 | 0.255 | 0.494 | 0.303 | 0.367 | 0.253 | 0.385 | 0.269 | 0.445 | 0.295 | 0.576 | 0.359 | 0.597 | 0.371 |
| | 192 | **0.358** | **0.255** | 0.359 | 0.263 | 0.513 | 0.310 | 0.382 | 0.259 | 0.395 | 0.273 | 0.461 | 0.302 | 0.610 | 0.380 | 0.607 | 0.382 |
| | 336 | **0.372** | **0.262** | 0.379 | 0.272 | 0.534 | 0.320 | 0.396 | 0.267 | 0.409 | 0.281 | 0.483 | 0.307 | 0.608 | 0.375 | 0.623 | 0.387 |
| | 720 | **0.416** | **0.284** | 0.417 | 0.291 | 0.597 | 0.346 | 0.433 | 0.287 | 0.449 | 0.305 | 0.527 | 0.310 | 0.621 | 0.375 | 0.639 | 0.395 |
| Solar | 96 | **0.165** | **0.231** | 0.170 | 0.246 | 0.183 | 0.232 | 0.185 | 0.246 | 0.191 | 0.257 | 0.190 | 0.243 | 0.214 | 0.311 | 0.316 | 0.369 |
| | 192 | **0.178** | **0.240** | 0.195 | 0.263 | 0.198 | 0.256 | 0.201 | 0.262 | 0.211 | 0.273 | 0.205 | 0.247 | 0.281 | 0.364 | 0.418 | 0.437 |
| | 336 | **0.192** | **0.252** | 0.217 | 0.282 | 0.205 | 0.261 | 0.209 | 0.266 | 0.228 | 0.285 | 0.219 | **0.250** | 0.294 | 0.378 | 0.438 | 0.467 |
| | 720 | **0.199** | **0.258** | 0.208 | 0.276 | 0.202 | 0.265 | 0.226 | 0.283 | 0.236 | 0.294 | 0.227 | 0.263 | 0.315 | 0.406 | 0.618 | 0.550 |
| Covid-19 | 12 | **0.892** | **0.041** | 0.998 | 0.046 | 3.584 | 0.075 | 1.236 | 0.054 | 2.643 | 0.087 | 6.505 | 0.110 | 7.607 | 0.316 | 7.695 | 0.406 |
| | 24 | **1.118** | **0.047** | 1.488 | 0.060 | 2.532 | 0.079 | 1.584 | 0.064 | 3.678 | 0.100 | 23.587 | 0.155 | 8.162 | 0.312 | 9.563 | 0.410 |
| | 48 | **2.090** | **0.072** | 2.505 | 0.082 | 12.922 | 0.140 | 2.639 | 0.089 | 5.836 | 0.131 | 33.467 | 0.206 | 9.458 | 0.328 | 9.563 | 0.440 |
| | 96 | **4.671** | **0.119** | 6.435 | 0.146 | 7.991 | 0.164 | 11.811 | 0.176 | 10.092 | 0.185 | 24.247 | 0.261 | 12.694 | 0.550 | 12.592 | 0.456 |
| PEMS | 12 | **0.059** | **0.161** | 0.064 | 0.167 | 0.091 | 0.202 | 0.063 | 0.166 | 0.078 | 0.187 | 0.094 | 0.204 | 0.283 | 0.394 | 0.584 | 0.607 |
| | 24 | **0.075** | **0.179** | 0.081 | 0.187 | 0.116 | 0.232 | 0.080 | 0.185 | 0.113 | 0.224 | 0.116 | 0.229 | 0.300 | 0.431 | 0.672 | 0.664 |
| | 48 | **0.102** | **0.206** | 0.111 | 0.215 | 0.165 | 0.271 | 0.109 | 0.213 | 0.167 | 0.274 | 0.147 | 0.255 | 0.396 | 0.476 | 0.879 | 0.781 |
| | 96 | **0.134** | **0.230** | 0.142 | 0.240 | 0.196 | 0.300 | 0.145 | 0.243 | 0.212 | 0.313 | 0.256 | 0.362 | 0.477 | 0.537 | 1.100 | 0.895 |

Table 3: Ablations on components. VST: Variable Scan along Time. TMB: Temporal Mamba Block. VAST: Variable-Aware Scan along Time. The average results of all predicted lengths are listed here.

| VST | TMB | VAST | ETTm2 | | Traffic | | Electricity | | Solar | |
|---|---|---|---|---|---|---|---|---|---|---|
| | | | MSE | MAE | MSE | MAE | MSE | MAE | MSE | MAE |
| ○ | ○ | ○ | 0.285 | 0.342 | 0.400 | 0.273 | 0.167 | 0.260 | 0.192 | 0.261 |
| ● | ○ | ○ | 0.284 | 0.341 | 0.383 | 0.268 | 0.164 | 0.260 | 0.197 | 0.266 |
| ○ | ● | ○ | 0.264 | 0.329 | 0.389 | 0.268 | 0.160 | 0.255 | 0.191 | 0.253 |
| ● | ● | ○ | 0.263 | 0.327 | 0.376 | 0.267 | 0.161 | 0.257 | 0.193 | 0.261 |
| ● | ● | ● | **0.262** | **0.325** | **0.373** | **0.262** | **0.155** | **0.251** | **0.184** | **0.247** |

## 6.2 ABLATION STUDIES AND ANALYSES

To validate the rationality and effectiveness of the proposed components, we conducted extensive ablation experiments as shown in Table 3, Table 4, and Figure 2.

**Component Ablation.** Table 3 presents the ablation on components. Initially, integrating VST with the baseline Mamba-based PatchTST led to performance improvements across most datasets, showcasing the benefits of considering all variables. Substituting solely with TMB resulted in a notable performance boost, underscoring the efficacy of TMB for temporal modeling. Combining VST and TMB yielded performance superior to VST alone but slightly below the original TMB, attributed to VST including all variables while TMB removes local bias causal convolutions, making it more sensitive to variable order. However, this issue is addressed by introducing VAST (see Table 3, row 4). Through these components, MambaTS achieves optimal performance.

**Dropout Ablation.** We further analyzed the role of the dropout in TMB. Figure 2 (left) shows the results of MambaTS with different dropout rates (0.1-0.5) on the Weather dataset. Compared to no dropout, the reduction in MSE of MambaTS increased with the dropout rate, achieving the best performance at 0.2 and 0.3, and performance degradation beyond 0.4. Figure 2 (right) illustrates

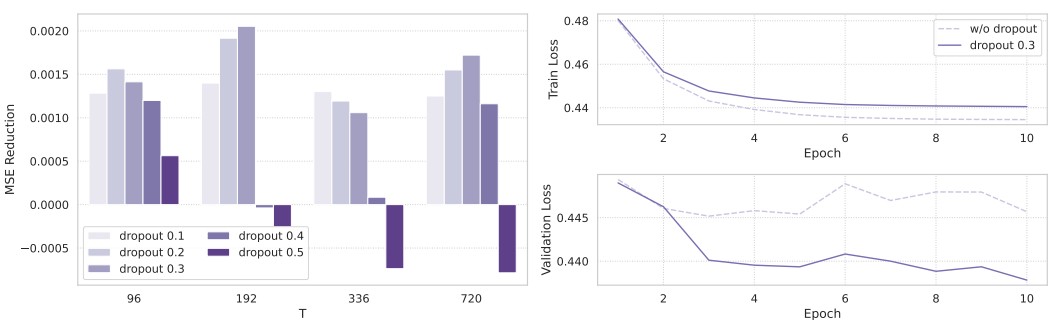

Figure 2: Dropout ablations of TMB. Left: TMB with varying dropout rates on Weather dataset. Right: Loss curves over training.

corresponding loss curves during training, indicating that the dropout in TMB helps prevent premature convergence and overfitting. Additionally, we observed that dropout facilitated lower validation loss.

**VAST Ablation.** In Table 4, we conducted extensive ablation studies on the VAST strategy, focusing on the design and selection of path decoding strategies. "W/o VPT" in Table 4 indicates that MambaTS was trained without VPT as the baseline. "Random (100x)" represents sampling 100 test runs after training with VPT and averaging the results. It can be observed that "Random (100x)" significantly outperformed "W/o VPT", which underscores the effectiveness of VPT. Further visual comparisons in Figure 3 show that even random variable scanning outperforms "W/o VPT" in most cases. We then explored different heuristic decoding strategies, including Greedy Strategy (GD), Local Search (LS), Lin and Lernighan (LK), and Simulated Annealing (SA). We defaulted to adopting SA as our solver. As a trade-off between efficiency and performance, we did not employ an exact ATSP solver due to its exponential complexity. As shown in Table 4 and Figure 3, SA consistently outperformed other solvers in terms of relative performance consistency.

| Scanning | ETTm2 | | Traffic | | Electricity | |
|---|---|---|---|---|---|---|
| | MSE | MAE | MSE | MAE | MSE | MAE |
| W/o VPT | 0.263 | 0.327 | 0.376 | 0.267 | 0.161 | 0.257 |
| Random (100x) | 0.262 | 0.326 | 0.374 | 0.265 | 0.158 | 0.256 |
| VAST (GD.) | 0.260 | 0.322 | 0.376 | 0.267 | 0.161 | 0.257 |
| VAST (LS.) | 0.261 | 0.325 | 0.375 | 0.265 | 0.157 | 0.254 |
| VAST (LK.) | 0.262 | 0.325 | 0.374 | 0.264 | 0.156 | 0.252 |
| VAST (SA.) | **0.259** | **0.321** | **0.373** | **0.262** | **0.156** | **0.251** |

Table 4: Ablations on VAST.

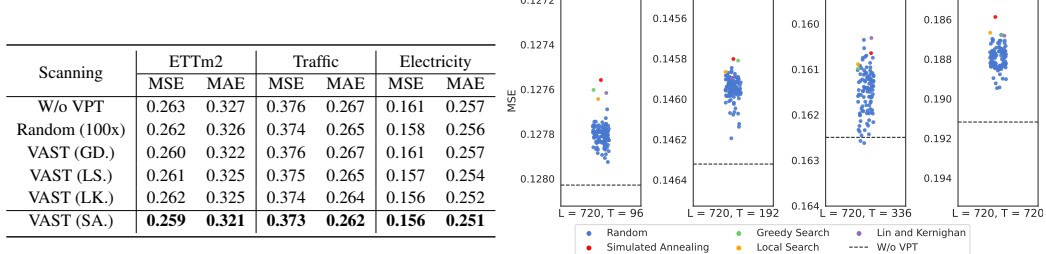

Figure 3: Sampling results of VAST. Employing swarm plot to prevent overlapping points (better in color).

### 6.3 MODEL ANALYSIS

**Increasing Lookback Window.** Previous studies have shown that Transformer-based methods may not necessarily benefit from a growing lookback window (Zeng et al., 2023; Nie et al., 2023), possibly due to distracted attention over the long input. In Figure 4, we assess MambaTS's performance in this context and compare it with several baselines. It can be observed that MambaTS consistently demonstrates the ability to benefit from the growing input sequence. iTransformer, PatchTST, and DLinear also show this benefit, but MambaTS's overall curve is lower than PatchTST and DLinear. Compared to iTransformer, MambaTS benefits more from a longer lookback window. Additionally, we notice that iTransformer seems to exhibit discontinuous gains on individual dataset tasks.

**Efficiency Analysis.** MambaTS integrates historical information from all variables using VST and conducts global dependency modeling through TMB. The computational complexity of MambaTS is

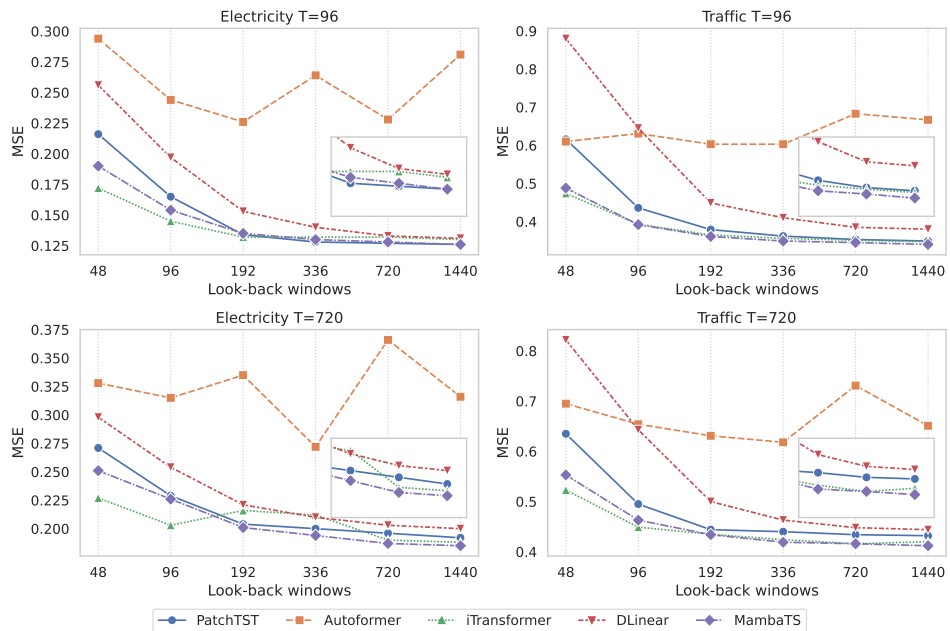

Figure 4: Performance of MambaTS in different datasets with varying length look-back windows.

$\mathcal{O}(\frac{KL}{P})$, where $K$ represents the number of variables, $L$ denotes the length of the lookback window, and $P$ signifies the patch stride. Table 5 outlines the computational complexity of other models.

Compared to the leading baselines, PatchTST and iTransformer exhibit complexities of $\mathcal{O}\left(\left(\frac{L}{P}\right)^2\right)$ and $\mathcal{O}(K^2)$, respectively. MambaTS effectively balances fine-grained temporal interactions with cross-variable dependency modeling. In our experiments, we typically set $L$ to 720 and $P$ to 48, leading to $M = \frac{L}{P} = 15$. For datasets with numerous variables, such as Traffic ($K = 862$), MambaTS's complexity is significantly lower than that of iTransformer, specifically $\mathcal{O}(KM) \ll \mathcal{O}(K^2)$. While variable independence-based methods like DLinear and PatchTST exhibit lower complexities, they often underperform by neglecting variable dependencies.

Table 5: Computational complexity analysis. SA: Self-attention. Conv: Convolution.

| Method | Temporal mixing | Variable mixing | Computational complexity |
|---|---|---|---|
| Autoformer | SA | MLP | $\mathcal{O}(L \log L)$ |
| FEDformer | SA | MLP | $\mathcal{O}(L)$ |
| MICN | Conv | Conv | $\mathcal{O}(K^2 L)$ |
| FourierGNN | GNN | GNN | $\mathcal{O}(KL)$ |
| DLinear | MLP | – | $\mathcal{O}(L)$ |
| PatchTST | SA | – | $\mathcal{O}\left(\left(\frac{L}{P}\right)^2\right)$ |
| iTransformer | MLP | SA | $\mathcal{O}(K^2)$ |
| MambaTS | TMB | TMB | $\mathcal{O}\left(\frac{KL}{P}\right)$ |

## 7 CONCLUSION

In this work, we present MambaTS, a innovative multivariate time series forecasting model built upon improved selective SSMs. We emphasize the essential role of causal graphs in LTSF, providing theoretical evidence for their ability to model global dependencies across time and variables through a single linear scan. To identify causal relationships, we propose Variable-Aware Scan along Time (VAST), which dynamically discovers variable interactions during training and utilizes an ATSP solver to determine the optimal variable scan order during inference. To avoid the entanglement of independent variables, we introduce the Temporal Mamba Block (TMB), which removes causal convolution from the original Mamba Block. Additionally, we incorporate dropout regularization for TMB's selective parameters to mitigate overfitting and enhance model performance. Gained from these insights and designs, MambaTS achieves global dependency modeling with linear complexity and establishes new SOTA results across multiple datasets and settings.

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
