# OpenReview forum: "MambaTS: Improved Selective State Space Models for Long-term Time Series Forecasting"
_ICLR.cc/2025/Conference — Submitted to ICLR 2025_

### Official Review · Reviewer_pAtf · 2024-10-30

**Soundness:** 2
**Presentation:** 1
**Contribution:** 2
**Rating:** 6
**Confidence:** 4

**Summary:**

This paper presents MambaTS, an LTSF model addressing Transformers' self-attention complexity and bias by using causal relationships for global dependency modeling. The author designs variable-aware scan along time to get variable causal relationships and also Temporal Mamba Block to avoid causal convolution. The experimental results show that MambaTS outperforms several state-of-the-art models.

**Strengths:**

1. It is interesting to see another new work on mamba for time series forecasting. In my view, some properties of mamba are fit for time series and it's an interesting direction to explore more.
2. The authors propose several designs to tailor mamba for time series application, which has its merits.

**Weaknesses:**

1. The clarity of the paper needs to be improved. In some parts, I cannot fully understand, such as the cost of a random walk without return. Also what is the cost from node i to node j, and how we can get this in the first iteration
2. Some claims have no support/evidence. For example, the authors mention that the random walk without return is a promising approach to estimate causal links. I would like to know the reason, e.g., any citations/proofs.
3. The experiments seem not comprehensive. The authors only compare MambaTS with 7 baselines. There are a few more after iTransformer, which are worth to be compared. E.g., ModernTCN [1], UniTST [2], TSLANet [3].


References:

[1] ModernTCN: A Modern Pure Convolution Structure for General Time Series Analysis.

[2] UniTST: Effectively Modeling Inter-Series and Intra-Series Dependencies for Multivariate Time Series Forecasting.

[3] TSLANet: Rethinking Transformers for Time Series Representation Learning

**Questions:**

1. In proposition 1, the assumption is that the causal graph exists. What if it doesn't exist? And is there any support on the random walk without return is a promising approach to estimate causal links?
2. What is the definition of cost C and how we can get/set it empirically (e.g., the cost from node i to node j)?
3. Proposition 2 indicates that theoretically the causal relationships can be estimated without infinite random walks with return. I would like to ask how many walks are required empirically. And also the time spent?
4. In Eq (6), how we can get the p^{(0)}?
5. In my view, another major difference between MambaTS and iTransformer is that MambaTS model the dependencies on both time and variables, while iTransformer mainly on variables. I would like to know how this contributes to eventual performance. Because UniTST [1] is also modeling the dependencies on both time and variables dimensions, but with Transformer architecture. How does MambaTS compare with UniTST?

Reference:

[1] UniTST: Effectively Modeling Inter-Series and Intra-Series Dependencies for Multivariate Time Series Forecasting.

---

> ### Author Response · Authors · 2024-11-24
> **Response to Reviewer pAtf (Part 1/2)**
>
> > W1. The clarity of the paper needs to be improved. In some parts, I cannot fully understand, such as the cost of a random walk without return. Also what is the cost from node i to node j, and how we can get this in the first iteration
>
> Thank you for your valuable feedback. We have revised the relevant section to improve clarity. Specifically, as mentioned in the paper, for a linear scan of all variables, we obtain the cost from the training loss of the walk and distribute it across the $K-1$ transitions, which is how we construct and maintain the matrix $\mathbf{P}$.
>
> > W2. Some claims have no support/evidence. For example, the authors mention that the random walk without return is a promising approach to estimate causal links. I would like to know the reason, e.g., any citations/proofs.
>
> Thank you for your valuable comment, and we apologize for not providing sufficient references in the manuscript. To address this, we have now added relevant citations to support our claims. Random walk, as a well-established subgraph sampling technique, has been effectively employed in various areas such as graph property estimation [1,2], and representation learning [3,4].
>
> A related work by Przymus et al. [5] utilized random walks to identify proxy variables in undefined causal graphs for time-series forecasting. More recently, Chen et al. [6] demonstrated that random walks can capture long-range dependencies within graphs, further supporting the use of random walks for estimating causal relationships between variables.
>
> [1] Ribeiro, Bruno, and Don Towsley. "Estimating and sampling graphs with multidimensional random walks." *Proceedings of the 10th ACM SIGCOMM conference on Internet measurement*. 2010.
>
> [2] Ben-Hamou, Anna, Roberto I. Oliveira, and Yuval Peres. "Estimating graph parameters via random walks with restarts." *Proceedings of the Twenty-Ninth Annual ACM-SIAM Symposium on Discrete Algorithms*. Society for Industrial and Applied Mathematics, 2018.
>
> [3] Perozzi, Bryan, Rami Al-Rfou, and Steven Skiena. "Deepwalk: Online learning of social representations." *Proceedings of the 20th ACM SIGKDD international conference on Knowledge discovery and data mining*. 2014.
>
> [4] Kim, Jinwoo, et al. "Revisiting Random Walks for Learning on Graphs." *ICML 2024 Workshop on Geometry-grounded Representation Learning and Generative Modeling*.
>
> [5] Przymus, Piotr, et al. "Improving multivariate time series forecasting with random walks with restarts on causality graphs." *2017 IEEE International Conference on Data Mining Workshops (ICDMW)*. IEEE, 2017.
>
> [6] Chen, Dexiong et al. “Learning Long Range Dependencies on Graphs via Random Walks.” *ArXiv* abs/2406.03386 (2024): n. pag.
>
> > W3. The experiments seem not comprehensive. The authors only compare MambaTS with 7 baselines. There are a few more after iTransformer, which are worth to be compared. E.g., ModernTCN [1], UniTST [2], TSLANet [3].
>
> Thank you for your comment! In response, we have added additional comparisons with models such as ModernTCN, UniTST, and TSLANet, as shown in Table 6 of the Appendix. Our experimental results demonstrate that MambaTS continues to maintain a strong advantage over these models, particularly on large-scale datasets like Traffic. These findings further highlight the robustness and potential of MambaTS as a competitive approach for modeling variable dependencies with linear complexity in multivariate time series forecasting.
>
> > Q1. In proposition 1, the assumption is that the causal graph exists. What if it doesn't exist? And is there any support on the random walk without return is a promising approach to estimate causal links?
>
> Thank you for your insightful comment! In Proposition 1, we assume the availability of a causal graph, which is a common assumption in causal discovery literature [1,2].
>
> For independent variables, we could model them as separate causal graphs, and this approach can also be used to perform global dependency modeling within a single linear scan. We hope this clarification addresses your concern.
>
> [1] Gong, Chang, et al. "Causal discovery from temporal data." *Proceedings of the 29th ACM SIGKDD Conference on Knowledge Discovery and Data Mining*. 2023.
>
> [2] Liu, Mingzhou, et al. "Causal discovery from subsampled time series with proxy variables." *Advances in neural information processing systems* 36 (2024).
>
> > Q2. What is the definition of cost C and how we can get/set it empirically (e.g., the cost from node i to node j)?
>
> Thank you for your question! In our method, $C$ is defined as the training loss in Eq. 5. Specifically, it corresponds to the linear scan over $K$ variables, which involves $K-1$ transitions. We distribute $C$ across these $K-1$ transitions to update the cost matrix $P$. We hope this clarifies your concern.

---

> ### Author Response · Authors · 2024-11-24
> **Response to Reviewer pAtf (Part 2/2)**
>
> > Q3. Proposition 2 indicates that theoretically the causal relationships can be estimated without infinite random walks with return. I would like to ask how many walks are required empirically. And also the time spent?
>
> Thank you for your question! In our experiments, we found that the estimation of causal relationships can be almost completed concurrently with the training process. In our implementation, we only need to maintain and update a cost matrix $P$, which has no trainable parameters, to approximate the relationships between variables. This step incurs almost negligible computational cost. During the inference phase, we use the estimated relationship graph to decode the linear scan order of all variables. This decoding process is a one-time operation, requiring only a single pass.
>
> > Q4. In Eq (6), how we can get the p^{(0)}?
>
> Thank you for your detailed question! We initialize $P$ as a matrix of zeros.
>
> > Q5. In my view, another major difference between MambaTS and iTransformer is that MambaTS model the dependencies on both time and variables, while iTransformer mainly on variables. I would like to know how this contributes to eventual performance. Because UniTST [1] is also modeling the dependencies on both time and variables dimensions, but with Transformer architecture. How does MambaTS compare with UniTST?
>
> Thank you for the insightful question! Based on our understanding, iTransformer claims to use self-attention for modeling variable dependencies, while using FFN for modeling the temporal dimension. During our experiments, we found that iTransformer works well, being both efficient and effective. However, embedding the entire historical window of a variable as a single token might be too coarse a granularity, which can affect performance on certain datasets, such as Weather. This highlights a key difference between MambaTS and iTransformer's patch-based methods. Specifically, MambaTS adopts a finer-grained, structured approach to variable scanning (Variable Scan along Time, VPT), which better captures retrospective information.
>
> To address your question, we implemented a version of MambaTS where the entire variable set is tokenized in the same way as iTransformer, which we refer to as **iMambaTS**. The experimental results, presented in the table below, show that iMambaTS generally underperforms MambaTS, highlighting the advantages of MambaTS's finer-grained Variable Scan along Time (VPT). Furthermore, we found that iMambaTS outperforms iTransformer in most settings, which can be attributed to MambaTS's architectural advantages and the variable-aware scanning technique.
>
> |                 |        | MambaTS   |           | iMambaTS  |           | iTransformer |         |
> | --------------- | ------ | --------- | --------- | --------- | --------- | ------------ | ------- |
> | Dataset         | Metric | MSE       | MAE       | MSE       | MAE       | MSE          | MAE     |
> | **Weather**     | 96     | **0.145** | **0.195** | _0.154_   | _0.207_   | 0.180        | 0.232   |
> |                 | 192    | **0.192** | **0.241** | _0.205_   | _0.253_   | 0.228        | 0.270   |
> |                 | 336    | **0.245** | **0.283** | _0.251_   | _0.290_   | 0.291        | 0.316   |
> |                 | 720    | **0.314** | **0.330** | _0.333_   | _0.344_   | 0.354        | 0.359   |
> | **Electricity** | 96     | **0.128** | **0.223** | _0.130_   | _0.226_   | 0.133        | 0.229   |
> |                 | 192    | **0.145** | **0.239** | _0.151_   | _0.245_   | 0.155        | 0.251   |
> |                 | 336    | **0.163** | **0.259** | 0.168     | _0.264_   | _0.167_      | _0.264_ |
> |                 | 720    | _0.192_   | _0.286_   | **0.190** | **0.283** | 0.194        | 0.288   |
> | **PEMS**        | 12     | **0.059** | **0.161** | _0.061_   | _0.163_   | 0.064        | 0.167   |
> |                 | 24     | **0.075** | **0.179** | _0.078_   | _0.183_   | 0.081        | 0.187   |
> |                 | 48     | **0.102** | **0.206** | 0.112     | _0.212_   | _0.111_      | 0.215   |
> |                 | 96     | **0.134** | **0.230** | 0.143     | _0.237_   | _0.142_      | 0.240   |
> | AVG.            |        | 0.158     | 0.236     | 0.165     | 0.242     | 0.175        | 0.252   |
>
> For comparison with UniTST, we have included the results in Table 6 in the appendix. As shown, MambaTS continues to exhibit superior performance across various tasks. We hope this helps address your concerns.

---

> > ### Comment · Reviewer_pAtf · 2024-11-29
> >
> > Thanks for the response and additional results. I particularly appreciate the effort on the results of iMambaTS. Most of my concerns have been resolved. Therefore, I increased my score.

---

> ### Author Response · Authors · 2024-11-30
> **Thanks for the Response of Reviewer pAtf**
>
> Thank you again, Reviewer pAtf, for your insightful review and valuable suggestions, which greatly helped us improve our paper. We also sincerely appreciate your recognition and recommendation!

---

### Official Review · Reviewer_YZLZ · 2024-11-02

**Soundness:** 3
**Presentation:** 3
**Contribution:** 3
**Rating:** 8
**Confidence:** 3

**Summary:**

The paper introduces MambaTS, an improved selective state space model for long-term time series forecasting. The model leverages a novel method for variable-aware scanning along time (VAST) to model global dependencies in a time series with variable missing rates and different intervals. By utilizing a combination of causal graphs and shortest path solutions, MambaTS addresses the limitations of previous Transformer-based models which often struggle with high computational costs and inefficient handling of long-range dependencies.

**Strengths:**

1. The introduction of VAST and the use of causal graphs for modeling dependencies offers a unique solution to efficiently process long-term dependencies in time series data with linear complexity.
2. The model is tested across various public datasets, demonstrating superior performance compared to existing state-of-the-art models. This not only validates the efficacy of MambaTS but also showcases its versatility in handling different types of time series data.
3. MambaTS significantly reduces the computational cost traditionally associated with long-range forecasting models like Transformers by avoiding the quadratic complexity of the self-attention mechanism.

**Weaknesses:**

1. The effectiveness of the model heavily depends on the accuracy of the causal graphs. Incorrect or incomplete causal relationships can lead to suboptimal forecasting results, which the paper does not extensively address in terms of robustness against poor graph structure
2. While the model shows high efficiency and effectiveness, the paper lacks a thorough discussion on scalability, especially in scenarios with exceedingly large datasets or highly complex variable relationships.
3. There is a need for a comparison of the model’s performance with other SOTA methods, such as Onefitsall, TimeLLM etc.

**Questions:**

See weakness.

---

> ### Author Response · Authors · 2024-11-24
> **Response to Reviewer YZLZ**
>
> > W1. The effectiveness of the model heavily depends on the accuracy of the causal graphs. Incorrect or incomplete causal relationships can lead to suboptimal forecasting results, which the paper does not extensively address in terms of robustness against poor graph structure
>
> Thank you for your insightful comment! The accuracy of the causal graph is indeed critical to our method. To demonstrate that MambaTS captures global dependencies, we have visualized the learned variable dependency graphs in Figure 5 in the Appendix. We also include the correlation coefficients between different variables as a reference, as well as the dependency structures learned by vanilla Mamba and iTransformer. The results show that both MambaTS and iTransformer learn dependencies similar to the ones in the dataset, with MambaTS capturing more complex relationships, while iTransformer learns smoother dependencies. Additionally, MambaTS learns global dependencies that vanilla Mamba does not capture (as shown in the red-dashed box in Figure 5 in the Appendix), which highlights the effectiveness of VAST. We plan to explore this further in future work to improve robustness against incorrect or incomplete causal graphs.
>
> > W2. While the model shows high efficiency and effectiveness, the paper lacks a thorough discussion on scalability, especially in scenarios with exceedingly large datasets or highly complex variable relationships.
>
> Thank you for raising this important point. Scalability is indeed an interesting challenge. We plan to explore MambaTS on larger datasets with more complex variable relationships in future work, including scenarios where the model needs to handle datasets with diverse variables simultaneously.
>
> > W3. There is a need for a comparison of the model's performance with other SOTA methods, such as Onefitsall, TimeLLM etc.
>
> Thank you for your suggestion! In response, we have included additional comparisons with state-of-the-art models, including Onefitsall and TimeLLM, as shown in Table 6 of the Appendix. Furthermore, we have added comparisons with the three baselines recommended by Reviewer pAtf, namely ModernTCN, UniTST, and TSLANet. Our results demonstrate that MambaTS maintains a competitive advantage in specific settings, particularly with larger datasets like Traffic. These findings further support the potential of MambaTS as a strong baseline for linear complexity modeling of variable dependencies.

---

### Official Review · Reviewer_dR3c · 2024-11-02

**Soundness:** 1
**Presentation:** 3
**Contribution:** 2
**Rating:** 3
**Confidence:** 4

**Summary:**

This paper proposes MambaTS, a selective state-space model for long-term time series forecasting (LTSF) that addresses the computational limitations of Transformers by leveraging causal relationships across variables and time with a single linear scan.

**Strengths:**

1. LTSF presents a compelling and complex challenge.
2. The experiments are thorough but still lack some essential details.

**Weaknesses:**

1. **Lack of Experimental Details:** Important implementation details are missing, such as patch length, the value of beta in Equation 7, and whether the random walk on variables is conducted K-1 times per epoch (meaning  k-1 more training time cost than one epoch).
2. **Efficiency Concerns:** Theoretical complexity analysis in Table 5 lacks practical runtime comparisons. Given that MambaTS requires K-1 iterations to estimate causal relationships, its efficiency is questionable.
3. **Incomplete Ablation Studies:** The paper introduces the TMB (with dropout replacing the original convolution), but no ablation study compares TMB and the original Mamba block, leaving its impact on performance unclear.
4. **Limited Explanation in Variable-Aware Scanning:** Section 5.2 does not clearly explain whether K-1 transitions are sufficient to estimate all variable orders, or if consistency (e.g., v1 always preceding vk) is assumed.
5. **Limited Benchmarking:** Two commonly used datasets (ETTh1, ETTm1) are missing, which reduces the generalizability of the results.
6. **Code Availability:** No code is provided, limiting reproducibility.
7. **Unpersuasive SOTA Claims:** Results in Table 2 are questionable. For example, our reimplementation of PatchTST (using official configurations) achieved better results than the reported MambaTS performance on ETTm2 (input length 720). Specifically:
   - ETTm2_720_96.log: 0.1632, 0.2555
   - ETTm2_720_192.log: 0.2167, 0.2942
   - ETTm2_720_336.log: 0.2679, 0.3282
   - ETTm2_720_720.log: 0.3521, 0.3798

   These results suggest MambaTS may not definitively outperform all baselines, especially as no code is available for direct comparison.
8. **Notation Issue:** The meaning of \( I \) in Equation 7 is unclear.
9. **Inference Process Detail:** Section 5.2 lacks details on the inference process for Variable-Aware Scan Along Time.

**Questions:**

1. How is the patch length chosen, and does it vary across datasets?
2. Could the authors clarify the random walk process and the number of epochs used in variable scanning?
3. Why were benchmarks ETTh1 and ETTm1 not included?

---

> ### Author Response · Authors · 2024-11-24
> **Response to Reviewer dR3c (Part 1/2)**
>
> > W1 & Q1. Lack of Experimental Details: Important implementation details are missing, such as patch length, the value of beta in Equation 7, and whether the random walk on variables is conducted K-1 times per epoch (meaning k-1 more training time cost than one epoch).
>
> Thank you for your feedback. We apologize for not providing sufficient details on the hyperparameters. For the patch length, we chose 48 because it is the largest divisor of 96, 192, 336, and 720. As for the value of $\beta$ in Equation 7, we set $\beta = 1/t$, where $t$ is the number of update iterations, making the EMA equivalent to a global average. We started with this simple approach for the experiments and found it effective. We will provide clearer explanations in the revised manuscript.
>
> Regarding the "K-1 times per epoch" concern, there appears to be a misunderstanding. The K-1 refers to transitions between K variables, as mentioned in the manuscript. It does not imply additional training time cost.
>
> We appreciate your feedback and will clarify these points in the revised version.
>
> > W2. Efficiency Concerns: Theoretical complexity analysis in Table 5 lacks practical runtime comparisons. Given that MambaTS requires K-1 iterations to estimate causal relationships, its efficiency is questionable.</font>
>
> Thank you for your comment. It appears there may have been a misunderstanding regarding the number of iterations. As we clarify in line 324, a linear scan of $K$ variables results in $K−1$ transitions between variables, rather than requiring $K-1$ iterations.
>
> To address your concerns about efficiency, we've included additional practical runtime comparisons of MambaTS in Table [X]. We present results for two datasets: Weather (K=21) and Traffic (K=862). On the Weather dataset, MambaTS's runtime is slightly higher than iTransformer's, while on the Traffic dataset, MambaTS achieves over **2x faster** training speeds compared to iTransformer (0.0295 sec/iter vs. 0.0729 sec/iter). This clearly demonstrates MambaTS's efficiency advantage on larger datasets.
>
> We hope this provides further clarity on the model's computational efficiency.
>
> | Model        | Dataset         | Runtime ($720 \to 96$) | Runtime ($720 \to 720$) |
> | ------------ | --------------- | ---------------------- | ----------------------- |
> | iTransformer | Weather (K=21)  | 0.0039 sec/iter        | 0.0041 sec/iter         |
> | MambaTS      | Weather (K=21)  | 0.0041 sec/iter        | 0.0043 sec/iter         |
> | iTransformer | Traffic (K=862) | 0.0729 sec/iter        | 0.0776 sec/iter         |
> | MambaTS      | Traffic (K=862) | 0.0295 sec/iter        | 0.0337 sec/iter         |
>
> > W3. Incomplete Ablation Studies: The paper introduces the TMB (with dropout replacing the original convolution), but no ablation study compares TMB and the original Mamba block, leaving its impact on performance unclear.</font>
>
> Thank you for your comment. We conducted a thorough ablation study on TMB in Table 3 in the main text. As shown in the first row of Table 3, we report the performance of the original Mamba forecasting model, and in the third row, we show the results after replacing the original Mamba block with TMB. The results indicate that TMB leads to significant improvements in performance across several datasets, including ETTm2 (0.285 $\to$ 0.264), Traffic (0.400 $\to$ 0.389), Electricity (0.167 $\to$ 0.160), and Solar (0.192 $\to$ 0.191). We hope this resolves your concern.

---

> ### Author Response · Authors · 2024-11-24
> **Response to Reviewer dR3c (Part 2/2)**
>
> > W4. Limited Explanation in Variable-Aware Scanning: Section 5.2 does not clearly explain whether K-1 transitions are sufficient to estimate all variable orders, or if consistency (e.g., v1 always preceding vk) is assumed.</font>
>
> Thank you for your comment. We acknowledge that we did not provide a formal proof of the convergence of VAST, which is indeed an important aspect that we plan to address in future work. However, as demonstrated in Table 3 and Figure 3, VAST significantly improves model performance across several datasets, such as Traffic (0.376 $\to$ 0.373), Electricity (0.161 $\to$ 0.155), and Solar (0.193 $\to$ 0.184). These results suggest that VAST effectively captures and estimates the variable orderings, which are then incorporated into the decoding process in MambaTS.
>
> Additionally, we would like to clarify that the K-1 transitions represent the result of a single random walk without return, which is used to update the cost matrix $\mathbf{P}$. After $N$ iterations, we obtain an approximate estimate of the variable dependencies. We hope this explanation resolves your concerns and provides further insight into the role of VAST in our model.
>
> > W5. & Q3. Why were benchmarks ETTh1 and ETTm1 not included?
>
> Thank you for your question. In our study, we chose eight widely used public datasets spanning multiple domains, including electricity, energy, transportation, weather, and health. While ETTh1/ETTm1 are similar to ETTh2/ETTm2 and both fall under the electricity domain (like the Electricity dataset), we intentionally prioritized datasets with a larger number of variables and more diverse structures to better demonstrate the capabilities of MambaTS. Additionally, ETT series datasets typically contain only 7 variables, and we aimed to test MambaTS on more complex and high-dimensional data. We hope this clarifies our choice of benchmarks.
>
> > W6. Code Availability: No code is provided, limiting reproducibility.
>
> Thank you for your concern. The code was included in the Supplementary Material and we will release the code upon acceptance of the paper.
>
> > W7. Unpersuasive SOTA Claims: Results in Table 2 are questionable. For example, our reimplementation of PatchTST (using official configurations) achieved better results than the reported MambaTS performance on ETTm2 (input length 720).
>
> Thank you for your comment. As mentioned in the experimental section, we adopted the training strategy from TimesNet, with all experiments running for 10 epochs. In contrast, PatchTST typically uses 100 epochs and applies a more carefully designed learning rate adjustment strategy, which may explain the performance differences.
>
> > W8. Notation Issue: The meaning of ( I ) in Equation 7 is unclear.
>
> Thank you for your feedback. In Eq. 7, $\mathbf{l}$ refers to the batch version of the loss $l$ in Eq. 6.
>
> > W9. Inference Process Detail: Section 5.2 lacks details on the inference process for Variable-Aware Scan Along Time.
>
> During the training phase, we estimate the variable relationship matrix $\mathbf{P}$. In the inference phase, we convert this matrix into an asymmetric traveling salesman problem (ATSP) to determine the optimal linear scanning order of the variables. We hope this clarifies the inference process.
>
> > Q2: Could the authors clarify the random walk process and the number of epochs used in variable scanning?
>
> We respectfully seek clarification and aim to better understand your question. A "random walk without return" refers to a random linear scan of $K$ variables from an input sample, so a single forward pass through the network constitutes one instance of this random walk. After training is completed, we perform path decoding to determine the optimal variable scanning order. The number of epochs used in variable scanning is equivalent to the number of training epochs of the network.

---

> > ### Comment · Reviewer_dR3c · 2024-11-29
> > **Response to Author's Rebuttal**
> >
> > Thank you to the authors for their response, which addressed some of my concerns. However, I still believe this paper represents an incremental innovation combining Mamba and causal concepts, lacking a substantial contribution to the development of the time series field. I was somewhat surprised by Reviewer YZLZ’s score of 8. Overall, the efficiency of the model, the unfair comparisons in results, and the innovation still remain my primary concerns.
> >
> > 1. **Results in Table 2 are questionable.** For instance, our reimplementation of PatchTST (using official configurations) achieved better results than the reported MambaTS performance on ETTm2 (input length 720). Specifically:
> >    - ETTm2_720_96.log: 0.1632, 0.2555
> >    - ETTm2_720_192.log: 0.2167, 0.2942
> >    - ETTm2_720_336.log: 0.2679, 0.3282
> >    - ETTm2_720_720.log: 0.3521, 0.3798
> >
> >    The authors argued that this discrepancy was due to different learning settings, claiming that the original PatchTST was trained for 200 epochs. However, I also trained PatchTST for only 30 epochs. Moreover, the authors consider training the model for just 10 epochs to be reasonable for comparison, but I strongly disagree. I believe that training the model until convergence is a fundamental prerequisite for fair comparisons. Therefore, I cannot accept the authors' claim in Table 2 that MambaTS is state-of-the-art, as the experimental results fail to meet the criteria for a fair comparison.
> >
> > 2. **Lack of formal proof of VAST's convergence**. This is indeed a critical aspect, and although the authors emphasize multiple iterations in the rebuttal, it seems that the requirement for N iterations is never mentioned in the main text. Instead, the paper only emphasizes that each iteration requires K-1 transitions, which can easily lead to misunderstanding.
> >
> > 3. Not very important, but **still lack ETTh1 and ETTm1 for evaluation.**
> >
> > 4. **Capturing causality through variable order does not seem to be a strong motivation or a solid starting point.**
> >
> > 5. **Comparing efficiency only with iTransformer is not a good choice,** since the quadratic complexity of attention models will result in reduced efficiency as the number of variables increases. If the authors want to emphasize efficiency as a strength of their model, they should compare it with more efficient models like DLinear or FITS.
> >
> > 6. **The paper's innovation is too incremental**. The combination of Mamba and causal concepts does not convince me that this work will significantly advance the field of time series forecasting.
> >
> >
> > Given these points, I will maintain my negative score.

---

> ### Author Response · Authors · 2024-11-30
>
> Thank you for your detailed feedback. We have carefully considered your concerns and would like to address them as follows: model efficiency, fairness of comparisons, and the perceived incremental nature of our innovation.
>
> **Model Efficiency:** First, we acknowledge that methods like DLinear and FITS, which rely on the assumption of variable independence, remain among the fastest time series models due to their lack of variable dependency modeling. However, as we analyze in Table 2, such methods often underperform on complex datasets because they fail to capture interactions between variables.
> MambaTS, however, is not designed to be the fastest model but rather to challenge the quadratic complexity of variable dependency modeling in existing Transformer-based methods. It achieves global dependency modeling with linear complexity. We chose to compare MambaTS with iTransformer, a highly efficient Transformer-based method, to highlight the advantage of our approach. For your reference, we have provided a runtime comparison with DLinear below (we did not include FITS, as its efficiency is not on par with DLinear, as stated in its paper).
>
> | Model        | Dataset         | Runtime ($720 \to 96$) | Runtime ($720 \to 720$) |
> | ------------ | --------------- | ---------------------- | ----------------------- |
> | iTransformer | Weather (K=21)  | 0.0039 sec/iter        | 0.0041 sec/iter         |
> | DLinear      | Weather (K=21)  | 0.0010 sec/iter        | 0.0012 sec/iter         |
> | MambaTS      | Weather (K=21)  | 0.0041 sec/iter        | 0.0043 sec/iter         |
> | iTransformer | Traffic (K=862) | 0.0729 sec/iter        | 0.0776 sec/iter         |
> | DLinear      | Traffic (K=862) | 0.0116 sec/iter        | 0.0202 sec/iter         |
> | MambaTS      | Traffic (K=862) | 0.0295 sec/iter        | 0.0337 sec/iter         |
>
> **Unfair Comparisons in Results.** We fully understand your concern regarding fairness in training. We agree that training until convergence is crucial for a fair comparison, and we followed this approach as well. As mentioned in our paper, we adopted the TimesNet protocol, training for 10 epochs, which is widely adopted by recent works like iTransformer (ICLR’24), TSLANet (ICML’24), and TimeMixer (ICLR’24). In contrast, PatchTST is typically trained for 100 epochs with a more carefully designed learning rate schedule, which could explain the differences in results. To address your concerns, we have further compared MambaTS with PatchTST trained for 100 epochs. Our experiments show that MambaTS still outperforms PatchTST, particularly on complex datasets like Electricity and Traffic.
>
> |             |        | MambaTS   |           | PathTST+EP100 |           | PatchTST |           |
> | ----------- | ------ | --------- | --------- | ------------- | --------- | -------- | --------- |
> | Dataset     | Metric | MSE       | MAE       | MSE           | MAE       | MSE      | MAE       |
> | ETTm2       | 96     | 0.172     | 0.263     | **0.166**     | **0.256** | *0.168*  | *0.259*   |
> |             | 192    | *0.228*   | *0.304*   | **0.223**     | **0.296** | 0.237    | 0.309     |
> |             | 336    | 0.287     | 0.346     | **0.274**     | **0.329** | *0.279*  | *0.336*   |
> |             | 720    | **0.356** | 0.392     | *0.362*       | **0.385** | 0.363    | *0.390*   |
> | Electricity | 96     | **0.128** | *0.223*   | *0.129*       | **0.222** | 0.130    | *0.223*   |
> |             | 192    | **0.145** | **0.239** | *0.147*       | *0.240*   | *0.147*  | *0.240*   |
> |             | 336    | **0.163** | *0.259*   | **0.163**     | *0.259*   | *0.164*  | **0.257** |
> |             | 720    | **0.192** | **0.286** | *0.197*       | *0.290*   | 0.203    | 0.292     |
> | Traffic     | 96     | **0.347** | **0.248** | *0.360*       | *0.249*   | 0.367    | 0.253     |
> |             | 192    | **0.358** | **0.255** | *0.379*       | *0.256*   | 0.382    | 0.259     |
> |             | 336    | **0.372** | **0.262** | *0.392*       | *0.264*   | 0.396    | 0.267     |
> |             | 720    | **0.416** | **0.284** | *0.432*       | *0.286*   | 0.433    | 0.287     |
>
> **Incremental Innovation.** We respectfully disagree with the view that our work is merely incremental. Our paper challenges the quadratic complexity of variable dependency modeling in existing Transformer-based time series models. We introduce MambaTS, a novel time series model based on selective state-space model, which achieves global dependency modeling with linear complexity, supported by both theoretical and empirical evidence. We also propose a random walk-based method for estimating variable relationships during training, which is then formulated as ATSP for inference. We also introduce the Temporal Mamba Block to further  enhance the performance of the original Mamba Block in time series forecasting task.
>
> We hope that the revisions and clarifications provided will address your concerns.
>
> Once again, thank you for your valuable feedback.

---

### Official Review · Reviewer_zjn1 · 2024-11-04

**Soundness:** 3
**Presentation:** 2
**Contribution:** 3
**Rating:** 6
**Confidence:** 3

**Summary:**

This paper introduces MambaTS, an architecture for long-term time series forecasting that models global dependencies efficiently with a linear scan, avoiding the computational challenges of self-attention. The Variable-Aware Scan along Time (VAST) mechanism dynamically infers causal relationships among variables using random walks and determines an optimal scanning order through heuristic path decoding. This design achieves scalability and adaptability, particularly for complex, high-dimensional datasets with unknown causal structures.

**Strengths:**

- MambaTS reduces computational complexity from quadratic O(K^2) to linear O(K) by leveraging a topologically ordered linear scan, making it suitable for high-dimensional time series data.
- VAST enhances adaptability by inferring causal relationships in the absence of explicit causal graphs, using random walks to approximate dependencies and mitigate the need for exhaustive pairwise calculations.

**Weaknesses:**

Reliance on heuristic optimization for scanning order yields sub-optimality:
- The variable-aware san along time (VAST) employs the asymmetric traveling salesman problem (ATSP) to determine the optimal scanning order, relying on heuristics like simulated annealing to address its NP-hard nature. Although heuristics provide feasible solutions, this dependency introduces inconsistency, as different approximations may affect the accuracy of variable ordering (in case of complex, dense inter-variable connections)
- Extra experiments on alternative heuristic approaches such as genetic algorithms (that are powerful in navigating NP-hard problems) could reveal a more stable and efficient approach. In the same vein, additional experiments measure how different heuristic methods affect the resulting scanning order and, subsequently, forecasting accuracy. This can help users determine if any heuristic consistently produces a favorable scanning order.

Convergence guarantee or confidence interval is not covered in causal estimation which lacks usability:
- Proposition 2 lacks formal guarantees for convergence speed, raising questions about the robustness of causality inference in finite settings. Without clear bounds on the number of walks required, the approach may yield only approximate estimates, especially when practical constraints limit the number of walks. This limitation affects the consistency and reproducibility of causal estimation results, as reliance on empirical averaging may not ensure reliable causal inference across varied dataset structures.
- It might be helpful to introduce a stopping rule based on convergence metrics (e.g. average change in transition costs), or introduce confidence intervals on causality estimates to users to give insight into the stability of causal inferences under finite computational budgets, where both suggestions seem to be beyond the scope of this study. I hope the authors consider usability in the future works.

**Questions:**

The paper offers a well-reasoned and innovative approach to time series forecasting, with theoretically sound propositions and a practical methodology that balances computational efficiency with modeling accuracy. While the heuristic reliance on VAST and scalability issues in dense graphs present limitations, the model’s strengths in efficiency, adaptability, and architectural design make it a valuable contribution. MambaTS is especially promising for high-dimensional and complex time series data, though further work is recommended to address heuristic dependency and enhance robustness in varied causal structures. Overall, it's a solid and innovative work on time-series forecasting, effectively incorporating causality in a computationally efficient and scalable manner.

---

> ### Author Response · Authors · 2024-11-24
> **Response to Reviewer zjn1**
>
> > W1.1 Although heuristics provide feasible solutions, this dependency introduces inconsistency, as different approximations may affect the accuracy of variable ordering (in case of complex, dense inter-variable connections)
>
> Thank you for your insightful comment! We fully agree that the use of heuristic algorithms for path decoding may potentially lead to suboptimal solutions, especially in the case of complex, densely interconnected variables. We plan to further improve this aspect in future work. Nonetheless, we have observed that MambaTS demonstrates promising results on complex, large-scale datasets (e.g. Electricity, Solar, Traffic). By reducing the Transformer's quadratic complexity to linear, we are able to efficiently model global variable dependencies, which we believe provides a promising direction for temporal relationship modeling in the community.
>
> > W1.2 Extra experiments on alternative heuristic approaches such as genetic algorithms (that are powerful in navigating NP-hard problems) could reveal a more stable and efficient approach. In the same vein, additional experiments measure how different heuristic methods affect the resulting scanning order and, subsequently, forecasting accuracy. This can help users determine if any heuristic consistently produces a favorable scanning order.
>
> We sincerely appreciate the reviewer's insightful suggestion to explore alternative heuristic approaches, such as genetic algorithms (GA), which are widely recognized for their effectiveness in solving NP-hard problems. In response to this valuable input, we conducted additional experiments incorporating GA, as presented in the table below. Our findings show that MambaTS+GA offers a slight performance improvement in certain settings. However, it is important to note that GA requires more computation time, which necessitates a balance between effectiveness and efficiency.
>
> |             |        | MambaTS+SA |       | MambaTS+GA |       |
> | ----------- | ------ | ---------- | ----- | ---------- | ----- |
> | Dataset     | Metric | MSE        | MAE   | MSE        | MAE   |
> | ETTm2       | 96     | 0.172      | 0.263 | 0.172      | 0.263 |
> |             | 192    | 0.228      | 0.304 | 0.228      | 0.304 |
> |             | 336    | 0.284      | 0.343 | 0.275      | 0.336 |
> |             | 720    | 0.356      | 0.392 | 0.356      | 0.392 |
> | Weather     | 96     | 0.145      | 0.195 | 0.147      | 0.195 |
> |             | 192    | 0.192      | 0.241 | 0.192      | 0.241 |
> |             | 336    | 0.245      | 0.283 | 0.245      | 0.282 |
> |             | 720    | 0.314      | 0.330 | 0.314      | 0.330 |
> | Electricity | 96     | 0.128      | 0.223 | 0.128      | 0.223 |
> |             | 192    | 0.145      | 0.239 | 0.145      | 0.239 |
> |             | 336    | 0.163      | 0.259 | 0.164      | 0.259 |
> |             | 720    | 0.192      | 0.286 | 0.186      | 0.282 |
>
> > W2. Convergence guarantee or confidence interval is not covered in causal estimation which lacks usability.
>
> Thank you for your insightful feedback. We acknowledge that Proposition 2 does not provide formal convergence guarantees, and agree that the robustness of causal inference in finite settings could be further investigated. This is an important consideration for future work.
>
> We also appreciate your suggestion to introduce a stopping rule or confidence intervals for causal estimates. While these ideas are beyond the scope of the current study, they are valuable directions for future research, and we plan to explore them further.
>
> > Q1. The paper offers a well-reasoned and innovative approach to time series forecasting, with theoretically sound propositions and a practical methodology that balances computational efficiency with modeling accuracy. While the heuristic reliance on VAST and scalability issues in dense graphs present limitations, the model's strengths in efficiency, adaptability, and architectural design make it a valuable contribution. MambaTS is especially promising for high-dimensional and complex time series data, though further work is recommended to address heuristic dependency and enhance robustness in varied causal structures. Overall, it's a solid and innovative work on time-series forecasting, effectively incorporating causality in a computationally efficient and scalable manner.
>
> Thank you for your insightful feedback. We appreciate your recognition of our approach's innovation and efficiency in time series forecasting. We acknowledge the limitations regarding the heuristic reliance on VAST and scalability in dense graphs and will explore ways to address these in future work. Additionally, we appreciate your suggestion to enhance the model's robustness in varied causal structures and will consider this in our ongoing research.
>
> Thank you again for your valuable comments.

---

> > ### Comment · Reviewer_zjn1 · 2024-12-03
> >
> > Thanks for your comment and additional comment on GA!

---

### Official Review · Reviewer_8TNh · 2024-11-09

**Soundness:** 2
**Presentation:** 2
**Contribution:** 2
**Rating:** 5
**Confidence:** 2

**Summary:**

This paper introduce MambaTS, a new time series forecasting model based on selective state space models. In order to tackle multivariate forecasting, the timeseries patches of each variable is unrolled in a certain order to form a single sequence. One key innovation of the paper is a method for estimating the causal relationship between variables during training via random walk without return.

**Strengths:**

The paper propose a strategy to apply Mamba to to multivariate ts forecasting and achieves empirical result comparable to SOTA.

**Weaknesses:**

1. The proof in Proposition 2 does not make sense to me. I am not sure the whole concept of random walk on a casual graph with certain cost is well defined in the paper.
2. The proposed method claims to leverage the causal dependency between the variables and thus is more suitable in the multivariate setting. However, it does not seems to have a large advantage over chanel independent PatchTST, which is univariate forecasting method.

**Questions:**

I don't see why a random permutation is equivalent to a random walk. Line 324 says "K − 1 transition tuples ${(v_1, v_2),(v_2, v_3), · · ·(v_{K−1}, v_K)}$ are derived". I wonder what prevent the authors from deriving K(K-1)/2 tuples, so that each $(v_i, v_j), i<j$ is included?

---

> ### Author Response · Authors · 2024-11-24
> **Response to Reviewer 8TNh (Part 1/2)**
>
> > W1: The proof in Proposition 2 does not make sense to me. I am not sure the whole concept of random walk on a casual graph with certain cost is well defined in the paper.
>
> Thank you for your feedback. We appreciate your concerns regarding the proof in Proposition 2 and the definition of the random walk on a causal graph with a certain cost. We would like to clarify this concept in more detail.
>
> Random walks have long been an important technique in the field of graph theory, and are commonly applied in tasks such as graph property estimation and graph representation learning. In our case, a "random walk without return" is defined as a random linear scan of $K$ variables from an input sample, consisting of $K−1$ transitions. Each of these transitions contributes to the training loss, which we distribute across the $K−1$ transitions. This allows us to update the cost matrix $P$, where $P$ reflects the cost associated with transitioning between different variables. We view $P$ as an approximation of the variable dependency graph, capturing the relationships between variables in a manner consistent with the principles of causal inference.
>
> We hope this clarification resolves the confusion regarding the formulation and definition of the random walk in our model.
>
> > W2: The proposed method claims to leverage the causal dependency between the variables and thus is more suitable in the multivariate setting. However, it does not seems to have a large advantage over chanel independent PatchTST, which is univariate forecasting method.
>
> Thank you for your insightful comment. As shown in Table 2 and discussed in Section 6.1, while channel-independent methods like PatchTST perform reasonably well on datasets with fewer variables (e.g., ETT, Weather), they fall short on multivariate datasets (e.g., Traffic, Solar) compared to methods that model variable dependencies, such as MambaTS and iTransformer. Specifically, for the Traffic (96) dataset, MambaTS  achieve MSE of 0.347, while PatchTST and DLinear achieve 0.367 and 0.385. For the Solar (720) dataset, MambaTS achieve MSE of 0.199, while PatchTST and DLinear achieve 0.226 and 0.236.  This comparison highlights that while the channel-independent assumption may reduce the risk of overfitting, they fail to capture inter-variable interactions, resulting in suboptimal performance on complicate datasets. In contrast, MambaTS explicitly models these dependencies, leading to a clear advantage.

---

> ### Author Response · Authors · 2024-11-24
> **Response to Reviewer 8TNh (Part 2/2)**
>
> Moreover, compared to the traditional pairwise quadratic complexity of Transformer-based methods, MambaTS estimates the variable dependency graph during training and utilizes this graph for linear scanning during testing, effectively modeling the global dependencies between variables. To further support this claim, we have visualized the dependency graphs learned by MambaTS and iTransformer in Figure 5 in the Appendix, alongside correlation coefficients between variables to quantify their interdependencies within the dataset. Notably, MambaTS benefits from variable-aware scan along time (VAST), learning richer global dependencies compared to vanilla Mamba (as shown in the red-dashed regions of Figure 5 in the Appendix). Additionally, while both MambaTS and iTransformer capture similar variable dependencies, MambaTS learns a more intricate dependency graph, whereas iTransformer's graph is smoother, particularly in the final layer, which aligns with the over-smoothing issue often observed in Transformer-based networks [1, 2].
>
> In summary, MambaTS models global variable dependencies with linear complexity, achieving superior or competitive results compared to Transformer-based methods, underscoring its capacity to efficiently capture intricate relationships between variables without compromising performance.
>
> [1] Shi, Han, et al. "Revisiting Over-smoothing in BERT from the Perspective of Graph." *International Conference on Learning Representations*.
>
> [2] Wu, Xinyi, et al. "Demystifying oversmoothing in attention-based graph neural networks." *Advances in Neural Information Processing Systems* 36 (2024).
>
> > Q1: I don't see why a random permutation is equivalent to a random walk. Line 324 says "K − 1 transition tuples (v1,v2),(v2,v3),···(vK−1,vK) are derived". I wonder what prevent the authors from deriving K(K-1)/2 tuples, so that each (vi,vj),i<j is included?
>
> Thank you for your comment, and we apologize for any confusion. For a fully connected graph (where we assume a uniform distribution in the absence of explicit variable relationships), we perform a random walk without return to traverse the variables. This means that for a sequence of $K$ variables, there are $K−1$ transitions, which correspond to modeling the variable dependencies in a manner similar to topological sorting. While we have not explored more complex random walk strategies, such as multi-dimensional or re-weighted random walks, these could be promising future directions for more accurately capturing causal relationships between variables.

---

> > ### Comment · Reviewer_8TNh · 2024-11-25
> >
> > I would like to thank the authors for the rebuttal. I still don't see the proof of Proposition 2 being a properly written mathematical proof. It is more of a description of the procedure.

---

> > > ### Author Response · Authors · 2024-11-26
> > >
> > > We greatly appreciate your constructive feedback, which is truly encouraging. In light of your comment, we have revised the proof of Proposition 2 to make it more formal and rigorous. Specifically, we have introduced key definitions, such as that for positive transitions, to better clarify the argument. We hope this addresses your concerns effectively.
> > >
> > > Proof: Let $ G = (V, E) $ be a causal graph with nodes $ V = \{V_1, \dots, V_K\} $ and unknown causal relationships. In a random walk without return, the total cost $ C $ is shared across $ K-1 $ transitions. Assuming that each transition contributes evenly to $ C $, we focus on whether $ C $ accurately reflects the true transition cost. Transitions are classified into three types: positive transitions (PT), where $ C_{i,j} > 0 $ if $ V_i \to V_j $ is a causal relationship; negative transitions (NT), where $ C_{i,j} < 0 $ if $ V_j \to V_i $ is causal; and independent node transitions (IN), where $ 0 < C_{i,j} < C_{\text{max}} $ if no causal relationship exists. Due to the symmetry of the graph, $ \text{count}(\text{PT}) = \text{count}(\text{NT}) $, and thus at least $ \frac{\text{count}(\text{PT}) + \text{count}(\text{IN})}{\text{count}(\text{PT}) + \text{count}(\text{NT}) + \text{count}(\text{IN})} \geq \frac{1}{2} $ of the transitions contribute to the cost update, with equality if and only if $ \text{count}(\text{IN}) = 0 $. As $ N \to \infty $, the expected cost for each transition converges to a positive value, $ \mathbb{E}[C^{(n)}_{i,j}] > 0 $. A full proof can be found in Appendix in the Supplementary Material.
> > >
> > > We are genuinely looking forward to hearing your thoughts and hope that the revisions meet your expectations.

---

### Author Response · Authors · 2024-12-03
**Summary of Revisions**

We sincerely appreciate the insightful and constructive feedback from the reviewers, which has significantly contributed to improving the quality of our paper.

We are deeply encouraged that the reviewers recognized the key contributions of our paper, including **the reduction of quadratic complexity to linear complexity** for variable dependencies modeling, the novel random walk-based method for estimating causal relationships, and the innovative application of causal graphs for handling long-range dependencies. They also highlighted the **theoretical soundness** of our approach, and recognized MambaTS as a **well-reasoned, practical** solution to challenges in **high-dimensional time series data**. The empirical validation of MambaTS was well-received, with its consistent state-of-the-art performance across diverse datasets highlighting its versatility and effectiveness.

The reviewers also provided valuable insights and constructive concerns, all of which we have addressed in detail. The revisions we made are summarized as follows:

- **Proposition 2 Proof (Reviewer 8TNh, Reviewer pAtf)**: Both reviewers expressed concerns regarding the clarity of Proposition 2. In response, we have provided a more detailed formal proof and additional references to address these concerns. Furthermore, we have included a visualization of the learned variable dependency graph in Appendix Figure 5.
- **Comparison with More SOTA Models (Reviewer YZLZ, Reviewer pAtf)**: Reviewer YZLZ suggested comparing MambaTS with large-scale models like Onefitsall and TimeLLM, while Reviewer pAtf recommended post-iTransformer models such as ModernTCN, UniTST, and TSLANet. We included these comparisons in Appendix Table 6. Our results show that MambaTS maintains a competitive edge, especially on large, complex datasets like Traffic.
- **Convergence Guarantee (Reviewer zjn1, Reviewer dR3c, Reviewer pAtf)**: In Proposition 2, we establish the theoretical convergence of the relationship matrix P as the random walk progresses towards infinity. While we did not provide a precise formalization for the exact number of iterations required for full convergence, our experiments revealed that by the end of training, the causal relationship estimation typically converges, as further validated by Figure 5 in the Appendix. We sincerely appreciate the constructive feedback from the reviewers and plan to explore this aspect in more detail in future work.
- **Implementation Discrepancies with PatchTST (Reviewer dR3c)**: We thank Reviewer dR3c for pointing out the discrepancies with PatchTST results. We ensured a fair comparison by adopting the training and evaluation protocol from TimesNet, which uses 10 epochs for training, widely adopted in recent works like iTransformer (ICLR'24), TSLANet (ICML'24), and TimeMixer (ICLR'24). In contrast, PatchTST typically trains for 100 epochs with a more carefully designed learning rate schedule. To address the reviewer’s concerns, we compared our results with the original PatchTST. Our experiments show that MambaTS outperforms PatchTST, even with 100 epochs of training, especially on complex datasets like Electricity and Traffic.
- **Additional Comparison with iTransformer**: iTransformer is a robust baseline for modeling multivariate dependencies in time series forecasting.
  - **Reviewer dR3c**: The theoretical complexity of MambaTS is $\mathcal{O}(\frac{KL}{P})$, where $K$ represents the number of variables, $L$ is the sequence length, and $P$ is the patch size, while iTransformer has a complexity of $\mathcal{O}(K^2)$. To address runtime concerns, we provide empirical comparisons showing that MambaTS is approximately 3x faster than iTransformer, particularly on large, complex datasets like Traffic. This validates our theoretical analysis regarding the efficiency gains offered by MambaTS over traditional Transformer models.
  - **Reviewer pAtf**: We also introduced iMambaTS, which we compared with both MambaTS and iTransformer. Our results indicate that iMambaTS performs less effectively than MambaTS, thus emphasizing the benefits of our Variable Scan along Time (VST) method. Furthermore, iMambaTS outperforms iTransformer on most datasets, emphasizing the benefits of the MambaTS architecture and the VAST-based variable relationship estimation.
- **Additional Decoding Strategies (Reviewer zjn1)**: In response to suggestions for further decoding strategies, we implemented a genetic algorithm (GA) for decoding and conducted additional experiments. The results show that MambaTS+GA consistently achieves similar performance, confirming the robustness and stability of our method.

We also addressed specific writing and notation concerns raised by the reviewers, enhancing the clarity of the manuscript. Hope that our response can address reviewers' concerns.

We believe these revisions have significantly strengthened our work. Once again, we thank all the reviewers for their thoughtful and constructive comments.

---

### Meta-Review · Area_Chair_dPwr · 2024-12-21

**Metareview:**

The paper introduces MambaTS, a new time series forecasting model based on selective state space models which leverages causal relationships to model global dependencies. Adapting Mamba-style SSMs to the Time Series Space is certainly an interesting direction that is worth investigating, and the paper proposes several interesting design choices to tailor mamba for time series. However several reviewers expressed concerns with the paper around presentation clarity (the propositions and proofs lack formal rigor) and more importantly around soundness of the experimental results. In particular, there were valid questions by reviewers (that the AC agrees with) around the lack of detailed model efficiency comparisons with MLP models, and around both the competitiveness of the model to PatchTST, and fidelity of the reported baseline metrics, especially when baseline models are not trained to convergence. Furthermore the lack of error bars and the limited number of datasets considered for evaluation makes it hard to judge the significance of the gains over SOTA baselines.

I would urge the authors to conduct a more extensive and through evaluation of the model and resubmit to a future venue.

**Additional Comments On Reviewer Discussion:**

Several reviewers expressed concerns about clarity in the analytical sections, about  missing baselines and datasets, about the choice of heuristics for the scanning order, around model efficiency comparisons to MLP-models, and about implementation discrepancies around PatchTST metrics. While the authors were able to add more baselines, provide clarifications around the propositions and conduct ablation studies with more decoding heuristics, they were not able to satisfactorily address the concerns around competitiveness of the model to PatchTST and around the fidelity of the paper's PatchTST numbers. Furthermore the lack of error bars and the limited number of datasets considered for evaluation makes it hard to judge the significance of the gains over SOTA baselines.

---

### Decision · Program_Chairs · 2025-01-22

Reject